# Dietary Phenolic Compounds: Their Health Benefits and Association with the Gut Microbiota

**DOI:** 10.3390/antiox12040880

**Published:** 2023-04-04

**Authors:** Yoko Matsumura, Masahiro Kitabatake, Shin-ichi Kayano, Toshihiro Ito

**Affiliations:** 1Department of Nutrition, Faculty of Health Sciences, Kio University, Kitakatsuragi-gun, Nara 635-0832, Japan; 2Department of Immunology, Nara Medical University, Kashihara, Nara 634-8521, Japan

**Keywords:** phenolic compounds, antioxidant, gut microbiota

## Abstract

Oxidative stress causes various diseases, such as type II diabetes and dyslipidemia, while antioxidants in foods may prevent a number of diseases and delay aging by exerting their effects in vivo. Phenolic compounds are phytochemicals such as flavonoids which consist of flavonols, flavones, flavanonols, flavanones, anthocyanidins, isoflavones, lignans, stilbenoids, curcuminoids, phenolic acids, and tannins. They have phenolic hydroxyl groups in their molecular structures. These compounds are present in most plants, are abundant in nature, and contribute to the bitterness and color of various foods. Dietary phenolic compounds, such as quercetin in onions and sesamin in sesame, exhibit antioxidant activity and help prevent cell aging and diseases. In addition, other kinds of compounds, such as tannins, have larger molecular weights, and many unexplained aspects still exist. The antioxidant activities of phenolic compounds may be beneficial for human health. On the other hand, metabolism by intestinal bacteria changes the structures of these compounds with antioxidant properties, and the resulting metabolites exert their effects in vivo. In recent years, it has become possible to analyze the composition of the intestinal microbiota. The augmentation of the intestinal microbiota by the intake of phenolic compounds has been implicated in disease prevention and symptom recovery. Furthermore, the “brain–gut axis”, which is a communication system between the gut microbiome and brain, is attracting increasing attention, and research has revealed that the gut microbiota and dietary phenolic compounds affect brain homeostasis. In this review, we discuss the usefulness of dietary phenolic compounds with antioxidant activities against some diseases, their biotransformation by the gut microbiota, the augmentation of the intestinal microflora, and their effects on the brain–gut axis.

## 1. Introduction

Phenolic compounds are components that contribute to the bitterness, astringency, and pigmentation of most plants. In addition to providing color to flowers, the physiological role of these compounds in plants is to confer biological protection against damage caused by ultraviolet rays, feeding by insects and herbivores, and pathogenic microorganisms. The type of phenolic compounds is dependent on its chemical structure [1] and includes well-known “catechins”, “isoflavones”, and “anthocyanins”. Phenolic compounds and their analogs have a wide variety of molecular sizes and structures (Figure 1). 

Previous studies on the antioxidant activity of phenolic compounds confirmed their role in the detoxification of excess reactive oxygen species (ROS) and the prevention of lifestyle-related diseases. The biological effects of phenolic compounds depend on the amount consumed and their digestion, absorption, and bioavailability. The majority of these compounds are not absorbed in the small intestine and reach the colon, in which glycosides are hydrolyzed and degraded by intestinal bacteria, generating various catabolites [2]. These catabolites have been found to contribute to human health. 

Among health issues, lifestyle diseases and neurodegenerative diseases are of great concern. As a dietary method that contributes to health, there is a ketogenic diet that mainly consists of lipids which is useful for Alzheimer’s disease relief [3] or prevention of obesity and diabetes [4]. In addition to these kinds of diet, dietary phenolic compounds and their catabolites also have health benefits in cardiovascular diseases [5], rheumatoid arthritis [6,7], depression [8,9], and eye diseases [10].

Research on intestinal bacteria has evolved in the past 20 years. The types and composition of bacteria that make up the intestinal flora may be investigated using a 16S rRNA-based metagenomic analysis. The type and composition of intestinal bacteria change under different disease states or with damage, which affects the regulation of metabolism and the immune system by these bacteria. In recent years, it has become possible to investigate the mechanisms by which the ingestion of phenolic compounds derived from various foods change the composition of intestinal bacteria and also their effects on the body. Dysbiosis of the intestinal microbiota is attracting attention as one of the pathogenic mechanisms of neurodegenerative diseases [11,12]. In the past decade, oxidative stress, inflammation, and impaired autophagy have been identified as pathogenetic factors for neurodegenerative diseases, such as Parkinson’s disease, Alzheimer’s disease, and amyotrophic lateral sclerosis [13,14,15,16]. Phenolic compounds, which are expected to exert antioxidant effects in vivo, may be involved in the attenuation or prevention of neurodegenerative diseases.

In our recent study in mice, administration of persimmon-derived tannin, a type of phenolic compound, suppressed the symptoms of *Mycobacterium Avium* Complex (MAC) infection [17], and decreased the severity of ulcerative colitis [18]. Furthermore, it is expected that persimmon-derived tannin is degraded by intestinal bacteria and the catabolites showed antioxidant activities in vivo [19]. In this review, we summarized the findings of studies in which the administration of phenolic compounds augmented the intestinal flora in vivo and exerted beneficial effects on health. Furthermore, we discussed some phenolic compounds that are indigestible and those with active substances that currently remain unknown.

## 2. Flavan-3-Ols

Flavan-3-ols (flavanols) are a group of flavonoids that have a 2-phenyl-3,4-dihydro-2*H*-chromen-3-ol skeleton. Dietary flavan-3-ols are abundant in cocoa, tea, apples, grapes (including red wine), berries, plums, apricots, and nuts. Flavan-3-ols are complex flavonoids in which monomers, such as catechins and epicatechins, make up units to form oligomers and polymers. They are components of proanthocyanidins, and many analogs exist in nature. Catechins, major dietary monomers, are abundant in tea leaves, and many studies have investigated their antioxidant properties [20,21]. Unlike other classes of flavonoids, flavan-3-ols are not present in a glycosylated form in foods [22] and monomeric flavan-3-ols are quickly absorbed in the small intestine. The galloylation and polymerization of flavan-3-ols were shown to significantly delay intestinal absorption [23]. Therefore, when oligomers and polymers reach the colon, they need to be metabolized by the colonic microbiota to provide health benefits. 

The mechanisms underlying the antioxidant effects of monomers have been reported [24]. The antioxidant capacity of flavan-3-ol monomers is exerted through phenolic hydroxyl groups that trap ROS and the chelation of iron ions to prevent lipid peroxidation [25,26]. By indirectly employing antioxidant pathways, flavan-3-ols regulate the synthesis of antioxidant-related enzymes and the signaling pathways of oxidative stress [27]. However, the mechanisms of action of oligomers and polymers remain unclear. 

### 2.1. Dietary Source and Metabolism of Flavan-3-Ols

#### 2.1.1. Tea

Tea is a major source of catechins. Various types of tea are available from the *Camellia sinensis* (L.) plant, depending on the harvesting and processing of its leaves. Green tea is unfermented tea; black tea is completely fermented tea; white tea and oolong tea are tea types with different degrees of fermentation [28]. There are five main types of catechins present in tea: (+)-catechin, (−)-epicatechin (EC), (−)-epigallocatechin (EGC), (−)-epicatechin gallate (ECG), and (−)-epigallocatechin gallate (EGCG) (Figure 2) [29,30]. EGCG is the most abundant catechin in unfermented teas (green tea and white tea) [31]. During the fermentation of black tea, catechins are oxidized by polyphenol oxidase to complex structures, such as theaflavin dimers and thearubigin polymers [32]. Tea phenolic compounds and their metabolites possess antibacterial properties against pathogenic bacteria, such as Clostridium perfringens, C. difficile, Escherichia coli, Salmonella, and Pseudomonas, and enhance the activities of probiotics, including Bifidobacterium and Lactobacillus species, thereby improving the overall balance of intestinal microbes [33,34]. The products of the intestinal bacterial catabolism of major tea catechins are shown in Figure 3 [35].

Theaflavins and theasinensins are catechin dimers that are not absorbed in the small intestine to the same extent as catechin; they reach the large intestine and are metabolized by intestinal bacteria enzymes [36,37,38]. Four theaflavins exist in black tea: theaflavin (TF), theaflavin-3-gallate (TF3G), theaflavin-3′-gallate (TF3′G), and theaflavin-3,3′-digallate (TFDG) (Figure 2), with TFDG being the most abundant [39]. TFDG alters the composition of the intestinal flora, similar to EGCG; however, the metabolic profile was significantly different [38]. The accumulation of further findings from in vivo studies is expected. Theasinensins are also catechin dimers with two galloyl groups; five theasinensins in fermented tea have been identified and named theasinensins A, B, C, D, and E (Figure 2) [40]. Theasinensin A is the most abundant among the five compounds [37]. The galloyl group is easily removed by intestinal bacteria and decomposed into theasinensin C. However, the progression of the subsequent reaction is slower than that of EGCG, and the whole picture remains unclear. In vivo studies are needed on these compounds, and the findings obtained will contribute to human health [37].

#### 2.1.2. Cocoa

Cocoa is generally produced by fermenting and roasting the seeds of *Theobroma cacao* and then pulverizing the cocoa cake obtained by removing the fat content. Although flavan-3-ols are relatively abundant in cocoa, its components vary depending on the type of cacao, place of origin, time of harvest, and processing of cocoa [41,42,43]. Cocoa flavan-3-ols, along with (+)-catechin and procyanidin B1 and B2 (Figure 4), as well as trace amounts of other flavanols [44], mostly exist as EC.

EC and procyanidin B1 in cocoa powder are metabolized in the intestines (Figure 5) [45]. Phenolic compounds in cocoa are metabolized in both the small and large intestine to produce metabolites that affect human health. [41,45,46]. 

### 2.2. Health Benefits of Flavan-3-Ols

#### 2.2.1. Tea 

A well-established causal relationship has been reported between the intake of EC and the regulation of cardiovascular function [47,48]. EC is rapidly absorbed, and its metabolites are excreted in the urine 72 h after consumption [49]. Although EC does not affect the composition of the microbial flora [50], EC phase II and gut microbiota metabolites may induce complex nutrigenomic/epigenomic changes that regulate the function of brain endothelial cells [49,51]. In other words, the metabolites of EC may reduce the risk of neurodegenerative diseases by maintaining the integrity of cerebrovascular endothelial cells, suggesting that the intake of EC contributes to improvements in cognitive ability [51].

The ingestion of tea reportedly attenuates alcoholic liver disease [52]. The administration of tea extract has been shown to activate antioxidant enzymes in the liver, change the intestinal flora, and promote liver function [53,54]. Although some types of teas promote liver function, others exert the opposite effects; therefore, further research on this subject is required [52,53]. 

EGCG is the major catechin found in unfermented tea [28] and exhibits the highest antioxidant activity among the four catechin monomers in vitro [30]. EGCG may attenuate non-alcoholic fatty liver disease (NAFLD) by regulating the interaction between the gut microbiota and bile acids [55]. 

NAFLD is closely associated with the gastrointestinal microflora and its dysbiosis [56,57]; therefore, further research on the treatment and prevention of NAFLD is needed. EGCG reportedly prevents the occurrence of NAFLD by regulating the intestinal flora. *Akkermansia muciniphila*, belonging to the phylum Verrucomicrobia, has been implicated in obesity, glucose metabolism, and intestinal immunity [58]. The abundance of the genus *Akkermansia* has been shown to increase with the intake of phenolic compounds and exerts anti-obesity effects [59]. Furthermore, EGCG intake increased the abundance of the genus *Akkermansia* in mice compared to a high-fat diet [55]. 

Inflammatory bowel disease (IBD) is an inflammatory disease that collectively refers to ulcerative colitis (UC) and Crohn’s disease, which are generally considered to have unknown (non-specific) etiologies. Catechins exhibit anti-inflammatory, antioxidant, and antibacterial activities, which may improve the abnormal condition of intestinal bacteria caused by IBD [60,61,62,63]. However, depending on the doses of catechin examined, conflicting findings have been reported; therefore, further research on this subject is needed [27].

Catechins in tea are metabolized into phenyl-γ-valerolactones by the action of intestinal bacteria as shown in Figure 3. Phenyl-γ-valerolactones regulate cellular proteolysis and exert neuroprotective effects [64]. In cell lines, EGCG, EGC, and ECG have been reported to inhibit amyloid-β-induced inflammation and neurotoxicity [65,66,67,68]. Animal studies also revealed the beneficial effects of EGCG on neurodegeneration in animal models of Alzheimer’s disease [69] and Parkinson’s disease [70,71]. Furthermore, EGCG was shown to affect hypoxia-induced neuroinflammation in cell lines [72]. Based on these findings, the intake of catechin may be effective against neurodegenerative diseases. However, there are many issues that need to be considered in clinical studies on humans, such as intake as food or supplements, dietary habits, and regional characteristics, and thus, further research is necessary.

#### 2.2.2. Cocoa

Cocoa powder has been shown to affect the gut microbiota by changing their metabolites and promoting the growth of *Lactobacillus* and *Bifidobacterium* groups in pigs [73] Flavanols in cocoa may function as prebiotics to maintain intestinal immunomodulation by regulating the gut microbiota [74,75,76]. The ingestion of cocoa powder was previously suggested to change the intestinal flora of the diabetic Zucker rat model, by strengthening the intestinal barrier and ameliorating colonic inflammation, thereby attenuating diabetes [77]. Cocoa powder was also shown to down-regulate inflammation markers and suppress inflammation-related colon carcinogenesis; therefore, its consumption may be promising for the prevention of intestinal inflammation and related cancers [78]. Cocoa flavanols also exert endothelium-dependent vasodilatory effects [79], suggesting their potential to ameliorate cardiovascular diseases [80].

Flavan-3-ols derived from cacao are metabolized into phenyl-γ-valerolactones by the action of intestinal bacteria, similar to the above-described tea catechins (Figure 5). Therefore, they may be effective against neurodegenerative diseases [81,82,83].

However, difficulties are associated with investigating the effects of cocoa flavan-3-ols in vivo due to the selection of an appropriate dose and their complex relationship with the intestinal flora [84]. Since cocoa powder also contains dietary fiber and alkaloids, such as theobromine, further studies on its effects on human health are warranted.

## 3. Condensed Tannins 

Tannin is a general term for astringent plant components that exist widely throughout the plant kingdom and have been traditionally used to tan leather. There are two types of tannins, one of which is hydrolyzed tannins which are polymers of ellagic acids or gallic acids, and the other is condensed tannins which are polymers of catechins. They are hydrolyzed or decomposed under specific conditions and produce low molecular weight phenolic compounds. The astringent skin of chestnuts and walnuts contain hydrolyzed tannins and astringent persimmons contain condensed tannins. Red wine also contains condensed tannins, but the degree of polymerization of catechins are altered depending on the degree of fermentation and the manufacturing method. In this chapter, we will focus on condensed tannins which are a component of astringent persimmon fruits.

### 3.1. Dietary Source and Metabolism of Tannins

#### Astringent Persimmon

Astringent persimmon fruits (*Diospyros kaki* Thunb.) contain large quantities of kaki tannin, a type of condensed tannin, such as EC, EGC, ECG, and EGCG (Figure 1) [85]. However, the structure of kaki tannin has not yet been clarified. Soluble kaki tannins in astringent persimmon fruits are converted into insoluble kaki tannins via dehydration, and dried persimmons lose their bitterness and have a sweet taste. Moreover, kaki tannins are reportedly non-hydrolyzable and non-digestible, but exhibit high antioxidant activity [86,87]. 

### 3.2. Health Benefits of Tannins

#### Astringent Persimmon 

Kaki tannin has the property of binding with bile acids [87] and the effect of lowering cholesterol and ameliorating glucose metabolism [88,89]. Kaki tannins have also been reported to reshape the gut microbiota in rats fed a high-cholesterol diet [90].

*Mycobacterium avium* complex (MAC) is the most common nontuberculous mycobacterium that causes chronic pulmonary infections in immunodeficient individuals. Kaki tannins, used as a dietary supplement, reduce the symptoms of pulmonary MAC infection [17], suggesting an impact on mucosal immune inflammation, including that of the gut, through their anti-inflammatory effects and changes to the gut microbial composition. Moreover, kaki tannins may need to be digested and/or fermented into smaller molecules in vivo prior to their absorption into the body in order to exert their beneficial effects. The artificial digestion of the non-extracted residues of dried persimmons containing kaki tannins suggested that intestinal bacteria degraded the tannins into lower molecular weight fragments [19]. 

UC is a chronic IBD induced by the dysregulation of the immune response in the intestinal mucosa. The pathogenesis of UC was less severe in a mouse model fed kaki tannins than in a control diet group [18]. Furthermore, the gene expression of an inflammatory cytokine (IL-1β) and chemokine (CXCL1) was significantly decreased in the tannin diet group. An analysis of the composition of the fecal microbiota of mice employing 16S ribosomal RNA gene sequencing revealed that a treatment with DSS significantly increased the abundance of the phylum *Enterobacteriaceae* in the control diet group, whereas it was significantly suppressed in the kaki tannin diet group.

Dietary supplementation with kaki tannins ameliorated the pathogenesis of MAC disease and DSS-induced colitis by suppressing the inflammatory response and changing the composition of the microbiota. However, further studies are needed to establish the optimal method of administration, select the appropriate concentration of kaki tannin, and elucidate the detailed chemical structures of the decomposed tannins. Although tannins have been shown to promote lipid metabolism in animal experiments [87,91,92], and similar findings were obtained for humans [93], the relationship between these findings and gut bacteria remains unclear. Therefore, human clinical trials are needed in the future to assess the health benefits of tannins.

## 4. Flavonols 

Flavonols, a subclass of flavonoids with a 3-hydroxyflavone skeleton, are widely present in plants [22]. Typical flavonols include myricetin (in grapes and berries), kaempferol (in tea, broccoli, and ginger), rutin (in asparagus and buckwheat), and quercetin (Figure 6). Quercetin is a representative flavonol that has been extensively examined and is present in vegetables and fruits, such as onions, broccoli, and apples. Flavonols generally exist in a glycosidic form and are deglycosylated and absorbed in the small intestine. After absorption, they are rapidly metabolized by phase II enzymes in the liver and circulate as methyl, glucuronide, and sulfate metabolites [94,95].

### 4.1. Dietary Sources and Metabolism of Flavonols

#### 4.1.1. Onions

Onions (*Allium cepa* L.) are used as an ingredient in various dishes. They are rich in flavanols, the most abundant of which is quercetin [96,97]. Quercetin (an aglycone) is mostly present in the outer skin and quercetin 4′-glucoside and quercetin 3,4′-diglucoside in the bulbs, which are generally edible [98,99]. Figure 7 shows the quercetin catabolites produced by intestinal bacteria and phase II enzymes in the liver. Quercetin derivatives in onions increase their bioavailability through cooking processes, such as baking, frying, and grilling [100].

#### 4.1.2. Buckwheat

Buckwheat is widely grown in Asia, Europe, and the Americas. Both common buckwheat (*Fagopyrum esculentum* Moench) and tartary buckwheat (*F. tataricum* (L.) Gilib.) are used as food sources, and the antioxidant activity of tartary buckwheat is higher than that of common buckwheat [101]. Rutin is the main flavonol in buckwheat, accounting for 90% of all phenolic compounds [102]. Rutin is a glycoside composed of flavonol aglycone quercetin along with disaccharide rutinose (Figure 6), and rutin is converted to quercetin by rutinosidase contained in seeds during grain milling [103]. Buckwheat is a potential gluten-free diet for people with gluten sensitivities and has been noted for its antioxidant properties and other health benefits [104].

### 4.2. Health Benefits of Flavonols

#### 4.2.1. Onions

Quercetin exhibits antioxidant, anti-inflammatory, and anti-osteoporotic activities [95,105]. The administration of quercetin and quercetin glycosides extracted from onion skin to rats on a high-fat diet increased serum antioxidant activity and significantly increased enzyme activity derived from intestinal bacteria [106]. In other words, quercetin effectively reduced the intestinal flora abnormalities induced by the high-fat diet. However, in human clinical studies, the administration of onion peel extracts to obese patients with hypertension did not attenuate their symptoms [107]. Similarly, in clinical studies on hypertension and rheumatoid arthritis, the administration of quercetin did not exert beneficial effects [108,109,110,111,112]. Based on the beneficial effects of onion peel observed in animal and cell culture experiments, clinical studies need to be performed on humans under various conditions, particularly obesity.

Quercetin glycosides are catabolized to produce phenolic acids by intestinal bacteria [113]. Among the phenolic acids derived from quercetin glycosides, 3,4-dihydroxyphenylacetic acid is the most effective at scavenging free radicals and inducing phase II enzymes [114]. Moreover, 3,4-dihydroxyphenylacetic acid significantly inhibits hydrogen peroxide-induced cytotoxicity [114,115]. Quercetin has been implicated in the attenuation of insulin resistance and atherosclerosis in obesity-related diseases [116,117,118,119]. It was found to promote intestinal homeostasis by changing the intestinal flora [120,121] and also plays a role in the prevention and treatment of inflammatory bowel disease [122,123]. 

A previous study demonstrated that quercetin and rutin effectively suppressed the aggregation of amyloid-β in cell lines, and thus, they are expected to be effective against Alzheimer’s disease [124]. Quercetin has potential in the treatment of Alzheimer’s disease in cell lines [125,126,127] and was effective in a mouse model of Alzheimer’s disease [128]. Therapeutic effects have been suggested in animal models of Parkinson’s disease, and quercetin may be effective against neurodegenerative diseases [129,130]. In addition, the combined use of quercetin and piperine (a type of alkaloid), which is a component of pepper, appeared to exert neuroprotective effects [131,132].

Although cell cultures and animal experiments have provided important findings, few clinical experiments have been conducted in humans to date; therefore, future research and verification are required.

#### 4.2.2. Buckwheat

Rutin and quercetin contained in tartary buckwheat regulate gut microbiota and are involved in lipid metabolism [133]. Rutin had little effect on attenuating obesity but tended to decrease fat deposition in the liver [133]. Phenolic compounds extracted from tartary buckwheat bran showed dose-dependent anticancer activity against human breast cancer MDA-MB-231 cells [134]. Further research is needed regarding the anticancer properties of rutin in humans [135]. It has been suggested that rutin has the potential to inhibit major proteases of SARS-CoV-2 in vitro [136,137]. 

Rutin and quercetin interact with buckwheat proteins and starch [138]. The presence of phenolic compounds such as rutin and quercetin reduces the digestibility of proteins and starches and allows them to be absorbed slowly [139,140]. While this is not a favorable outcome in terms of natural nutrient uptake, it also has some desirable consequences related to diabetes and lipid metabolism [141,142,143]. Concerning cardio-metabolic disease, meta-analyses have not yet yielded consistent results regarding the usefulness of phenolic compounds, such as rutin [144]. Recent studies suggest that buckwheat has inhibitory effects on Alzheimer’s disease and other neurological disorders [145], but it is not yet clear whether rutin is responsible for this effect [146]. Therefore, further research is needed.

## 5. Isoflavones

Isoflavones are flavonoids with 3-phenylchromone as the basic skeleton (Figure 8). They are abundant in plants of the legume family (Fabaceae), such as soybeans and kudzu. Isoflavones bind to estrogen receptors in the body and exert a number of effects because their chemical structures are similar to estrogen [147]. They may be beneficial, but also detrimental [148]. For example, while isoflavones are expected to effectively prevent osteoporosis, breast cancer, and prostate cancer, they also increase the risk of the onset and recurrence of breast cancer [148]. Glycosides are not easily absorbed in the small intestine and must be converted into aglycones, such as genistein and daidzein, to function in vivo [149,150].

### 5.1. Dietary Source and Metabolism of Isoflavones

#### Soybeans

Soybeans (*Glycine max* (L.) Merr.) are the most abundant source of isoflavones [151]. Many isoflavones, such as genistin and daidzin, are present in food (Figure 8). In the small intestine, lactase-phlorizin hydrolase and cytosolic β-glucosidase hydrolyze monoglucuronides to form aglycones [152,153]. The absorbed isoflavone aglycones are mainly metabolized to glucuronides and sulfates by endogenous phase I and phase II enzymes. Isoflavones are excreted into the intestines via the enterohepatic circulation, and unabsorbed isoflavones reach the colon and are metabolized to form the metabolite, equol, and other metabolites by intestinal bacteria [154] (Figure 9). Numerous studies have identified equol-producing bacteria; however, findings on the production of equol have been inconsistent because it is markedly affected by the diet of the host [154]. A previous study reported that 25–30% of the Western population possessed equol-producing gut bacteria, whereas they were detected in 50–60% of the Asian population [155]. 

### 5.2. Health Benefits of Isoflavones

#### Soybeans

Soybeans are rich in isoflavones, particularly genistin and daidzin [151]. Isoflavones are phytoestrogens, such as the female hormone 17-β-estradiol, which are less active than hormones, but exhibit estrogenic activity [156]. Therefore, the intake of isoflavones is expected to alleviate menopausal symptoms in women, increase bone formation, and reduce the incidence of cardiovascular disease. Equol is a metabolite of daidzin/daidzein formed by intestinal bacteria (Figure 9). It is more stable and more easily absorbed than daidzein [157] and exhibits stronger estrogenic activity than other isoflavones and isoflavone-derived metabolites [158,159,160,161]. Isoflavone aglycones and glycosides are both catabolized by enzymes of the intestinal microbiota to produce high levels of antioxidant substances, such as equol. A correlation has been reported between soybean intake and the attenuation of menopausal symptoms [162]. 

The intake of soy isoflavones has been suggested to reduce bone resorption, prevent some types of cancers, and improve learning [163,164,165,166]. These health effects are attributed to equol produced from soy isoflavones by the action of the intestinal microbiota. Therefore, these effects may be observed in individuals who produce equol in their intestines. Furthermore, the human gut microbiome is highly individualized, and its effects are inconsistent. This inconsistency poses a major challenge when considering the effects of isoflavones on humans. Adverse effects associated with the intake of soy isoflavones, including endometriosis, dysmenorrhea, and secondary infertility, have also been reported, and symptoms were ameliorated by the discontinuation of intake [167].

Isoflavones and their metabolites exert their effects by binding to the estrogen receptor (ER) and transmitting cell signals. However, isoflavones are agonists that activate ER as well as antagonists that inhibit it, which modulates estrogen signaling. Therefore, they may act as an endocrine disruptor, with more than just beneficial effects [167].

Animal studies showed that genistein, a soy isoflavone, was effective for the treatment of neurodegenerative diseases, such as Alzheimer’s disease [168,169,170] and Parkinson’s disease [171]. Early oral genistein therapy appeared to ameliorate the severity of disease in multiple sclerosis model mice [172].

In the future, we anticipate further advances in this field that will verify the effects of isoflavones and their metabolites on humans.

## 6. Phenylpropanoids

Phenylpropanoids, also called lignoids, are compounds that have a C6-C3 skeleton with a C3 group attached to an aromatic ring. Monomers include caffeic acid, which is widely present in plants, and chlorogenic acid (an ester of caffeic and quinic acid), which is abundant in green coffee beans. Sesamin is a dimer, also known as lignan, and is abundant in sesame seeds. Chlorogenic acid may be ingested from food. Figure 10 shows the chemical structures of the major phenylpropanoids.

### 6.1. Dietary Source and Metabolism of Phenylpropanoids

#### 6.1.1. Coffee

Coffee is one of the most consumed beverages in the world. It contains at least 30 types of chlorogenic acids [173]. The term “chlorogenic acids” refers to a group of phenolic compounds, of which approximately 400 have been discovered to date [174]. 5-*O*-caffeoylquinic acid is the main chlorogenic acid found in green coffee beans. Although the type and concentration of chlorogenic acids vary depending on the type of coffee bean, the roasting process, and extraction method, the beneficial health effects of coffee are related to its chlorogenic acid content, whether green or roasted. The high antioxidant activity of coffee is attributed to the amount of chlorogenic acid present [175]. Figure 11 shows the main chlorogenic acids found in coffee [176]. Approximately 30% of these chlorogenic acids are absorbed in the stomach or small intestine, while the remainder are transferred to the large intestine, in which they are metabolized into dihydroferulic acid, its 4-*O*-sulfate, and dihydrocaffeic acid-3-*O*-sulfate by intestinal bacteria [177,178].

#### 6.1.2. Sesame 

Sesame (*Sesamum indicum* L.) is an edible seed and source of high-quality edible oil. Sesame oil exhibits antioxidant activity and possesses health-promoting properties because it contains vitamin E and lignans [179,180]. The major lignans in sesame are sesamin and sesamolin, which are formed by the dimerization of two phenylpropanoids [181]. Sesamin and sesamolin exhibit weak antioxidant activities in vitro because they do not have phenolic hydroxyl groups [182]; however, they possess antioxidant properties after being metabolized in vivo to form hydroxyl groups [183,184] (Figure 12).

### 6.2. Health Benefits of Phenylpropanoids

#### 6.2.1. Coffee 

Chlorogenic acids exhibit antioxidant activity [185,186,187] and anti-obesity activity in vivo [188,189,190]. Daily coffee consumption reduces the risk of type 2 diabetes [191]. Chlorogenic acid from coffee possesses prebiotic properties in vivo [192,193]. Therefore, the daily consumption of coffee may contribute to the prevention of obesity and lifestyle-related diseases.

Coffee consumption has been suggested to reduce the risk of developing neurodegenerative diseases, such as Alzheimer’s disease, Parkinson’s disease, and dementia; however, coffee contains a wide variety of components and their interactions need to be investigated [194]. Since chlorogenic acid was shown to exert neuroprotective effects against Parkinson’s disease [195,196,197] and Alzheimer’s disease [198] in animal experiments, it is expected to exert similar effects in humans. More data needs to be collected because the bioavailability of active ingredients markedly varies between individuals.

#### 6.2.2. Sesame 

The lignans in sesame have a number of health benefits, including anticancer activity, reducing the risk of cardiovascular diseases, and anti-inflammatory effects [199,200]. They are converted into enterolignans by intestinal bacteria and exert their effects as phytoestrogens [201]. Sesame lignans have been shown to inhibit L-tryptophan indole-lyase (TIL) produced by intestinal bacteria and suppress the production of indoxyl sulfate, a uremic toxin, catalyzed by TIL [202]. The inhibition of TIL by sesame lignans has potential as a strategy to prevent and treat chronic kidney diseases. Although sesaminol triglucoside, a sesame lignan glycoside, did not inhibit TIL, it induced significant increases in *Lactobacillus* and *Bifidobacterium* and changed the intestinal microbial environment [203]. Sesamin may also augment the intestinal environment by increasing the abundance of beneficial genera of bacteria, including *Lactobacillus* and *Bifidobacterium*, in the intestinal flora [204]. Moreover, sesamin reportedly promoted the adhesion of epithelial colonocytes and probiotics [204]. 

Sesamin, sesamolin, and sesamol exert neuroprotective effects and are expected to be effective against neurodegenerative diseases, such as Alzheimer’s disease, Parkinson’s disease, and Huntington’s disease [205,206,207,208,209]. Sesamin and sesamolin are phenylpropanoid dimers, as shown in Figure 12, which differ in structure from the phenylpropanoid monomer sesamol. Sesamin and sesamolin have both been shown to reduce amyloid-β toxicity, whereas sesamol did not [209]. However, sesamol ameliorated scopolamine-induced cholinergic disorders [205], remodeled the intestinal microbiota, significantly increased the content of short-chain fatty acids, and attenuated cognitive deficits [206]. Although structure–activity relationships warrant further investigation, these sesame lignans have neuroprotective potential.

Collectively, these findings support the potential of sesame lignans to contribute to human health; however, only a few studies have been conducted in this area of clinical research.

## 7. Stilbenoids

Stilbenoids are derivatives of stilbene, an aromatic hydrocarbon called 1,2-diphenylethene. Major stilbenoids are shown in Figure 13. Resveratrol is a type of stilbenoid that is present in many plant food materials, such as grapes, cranberries, red currants, and peanut skin, as well as in their processed products [210]. As a stilbenoid phenolic compound, resveratrol has been extensively studied.

### 7.1. Dietary Source and Metabolism of Stilbenoids

#### Grapes and Wine

Resveratrol, a stilbenoid found in many plants, possesses antifungal and antibacterial properties. The food sources that contain resveratrol are grapes, wine [210], and grape seed oil [211]. Resveratrol, in its native state, is present at low amounts in humans, with only 1–8% being detected in serum. Although 75% is absorbed, it is rapidly metabolized [212,213]. Resveratrol undergoes glucuronidation and sulfation in the liver and duodenum to form resveratrol-3-glucuronide (R3G) and resveratrol-3-sulfate (R3S), respectively [214,215] (Figure 14). Moreover, the intestinal flora metabolizes resveratrol to dihydroresveratrol (DHR); however, this metabolism differs among individuals [216]. Resveratrol also crosses the blood–brain barrier due to the absence of phenolic degradation products by intestinal bacteria [217]. Therefore, resveratrol may suppress neurodegeneration in the central nervous system [218], and many studies have investigated its effects on the nervous system.

### 7.2. Health Benefits of Stilbenoids

#### Grapes and Wine

Moderate wine consumption has been suggested to exert beneficial effects on health. This is commonly known as “the French paradox” because of the low incidence of coronary artery disease despite the consumption of high saturated fats by the French population [219,220]. 

Resveratrol has been shown to modulate and promote intestinal barrier function in mice, suggesting its potential to augment the intestinal flora [221,222]. Resveratrol prevented obesity and attenuated NAFLD and NASH by modulating the intestinal flora, maintaining intestinal barrier integrity, and suppressing intestinal inflammation in animal models [223,224,225,226]. Furthermore, the administration of resveratrol reportedly affected the intestinal flora and steroid metabolism in middle-aged men with metabolic syndrome [214,227,228,229]; however, the underlying mechanisms have not yet been elucidated. Red wine consumption reduced the risk of coronary heart disease and prevented obesity through the beneficial effects of phenolic compounds in red wine, particularly resveratrol [230,231]. Moreover, as reported in animal studies, resveratrol augmented the intestinal flora; however, further research is needed to confirm its effects in humans. Resveratrol also functions as a phytoestrogen, suggesting that its effects differ in males and females. Resveratrol may be used to treat diabetic complications during pregnancy, endometriosis, and dysmenorrhea [232].

Animal models using grape seed oil have demonstrated wound healing activity [233,234], efficacy against ulcerative colitis [235], protection against carbon tetrachloride-induced liver inflammation [236]. In cell lines, pancreatic β-cell apoptosis induced by hyperglycemia was reduced [237]. In human clinical trials, a milky lotion containing grapeseed oil was found to be effective in treating skin problems on the cheeks [238], and the use of grapeseed oil as massage oil was effective in reducing the physiological edema of pregnancy [239]. Oral administration of grape seed oil suppressed serum triglycerides in humans [240].

The protective effects of resveratrol against neurodegeneration have been extensively examined in cell lines and animals. It may also play a role in the treatment and prevention of Alzheimer’s disease [241,242,243,244], Parkinson’s disease [245,246,247], Huntington’s disease [248], multiple sclerosis [249], and amyotrophic lateral sclerosis [250]. However, it has also been suggested to exacerbate multiple sclerosis [251].

## 8. Curcuminoids

Curcuminoids are lipophilic phenolic compounds with a diarylheptanoid structure and are the yellow pigment components of turmeric.

### 8.1. Dietary Source and Metabolism of Curcuminoids

#### Turmeric

Turmeric is a spice prepared from the underground stems of *Curcuma longa* L. It contains curcuminoids, such as curcumin, demethoxycurcumin, and bisdemethoxy-curcumin (Figure 15). Curcumin is the most abundant curcuminoid in turmeric [252] and contains phenolic hydroxyl groups in its chemical structure; therefore, it functions as a potent antioxidant that suppresses the production of ROS [253]. 

Due to its insolubility in water, curcumin is poorly absorbed in the gastrointestinal tract and thus, has low bioavailability [254]. It reaches the large intestine and is biotransformed, as shown in Figure 16, by phase I and phase II enzymes and enzymes derived from intestinal bacteria. The resulting metabolites exhibit anti-inflammatory and antioxidant activities [255,256].

### 8.2. Health Benefits of Curcuminoids

#### Turmeric

Although turmeric is used as a spice in many dishes, its consumption per person is low. Many human clinical trials have examined the effects of curcumin supplements. Since the amount of curcumin consumed may be an important factor, the accumulation of further findings is necessary. 

Curcumin exhibits anti-inflammatory, antibacterial, and anti-tumor activities [257,258,259,260,261] and also interferes with cancer-associated signaling pathways by targeting proteins and modulating gene expression [262,263]. In human clinical trials, the administration of curcumin capsules to patients with colorectal cancer reduced inflammation and oxidative stress in malignant colorectal epithelial cells. It also attenuated inflammation in patients with UC and gastrointestinal disorders [264,265,266,267].

Recent studies on curcumin and intestinal bacteria in animals reported that curcumin reduced cholesterol levels [268], ameliorated the pathology of UC [269,270], and promoted a favorable response to acute myeloid leukemia drugs [271]. Metabolites produced by the actions of intestinal bacteria may be responsible for these effects, and, in some cases, they may also be attributed to changes in the diversity of intestinal bacteria and flora. However, these effects were not observed under some conditions, and thus, further research is required to elucidate the underlying mechanisms [272]. Curcumin was previously shown to be effective against neurodegenerative diseases in many cell lines and animal studies [273,274,275,276,277]. It is also undergoing clinical trials for depression. Although curcumin may be useful in the treatment of depression, the confirmation of its therapeutic efficacy requires a multi-mechanistic approach due to the pathophysiological complexity of depression [278,279].

## 9. Other Phenolic Compounds: Dietary Sources, Metabolism, and Health Benefits

### 9.1. Protocatechuic Acid

Protocatechuic acid, a ubiquitous natural phenolic compound in plants, exerts diverse pharmacological effects, including antioxidant, antibacterial, antiviral, anticancer, anti-inflammatory, anti-aging, and anti-arteriosclerotic activities [280,281]. Protocatechuic acid is found not only in fruits and vegetables, but also in the herbal medicine Duzhong (*Eucommia ulmoides* Oliv.) [282]. Protocatechuic acid is also contained in oregano, which is used as a type of spice. After its ingestion, protocatechuic acid is absorbed through the intestinal epithelium, sulfated or glucuronylated through conjugation processes by phase II enzymes primarily in the liver, and then circulated throughout the body [283,284]. Protocatechuic acid is also produced in vivo as a metabolite via the degradation of phenolic compounds, particularly flavonoids, by the intestinal flora [285]. Figure 17 shows the degradation pathway of the production of protocatechuic acid from cyanidin [286]. 

Protocatechuic acid, a metabolite of various phenolic compounds, regulates oxidative stress and inflammatory responses. Furthermore, protocatechuic acid increases the energy expenditure of brown adipose tissue, which may reduce NAFLD [287], acts as an antidepressant [288], and inhibits the progression of neurodegenerative diseases, such as Alzheimer’s disease and Parkinson’s disease [286]. In addition, protocatechuic acid has been shown to affect the diversity and composition of the gut microbiota [286]. However, most of these findings were obtained from animal studies or cell culture experiments. Very few clinical trials have been conducted to date. Therefore, further animal experiments and clinical trials are required to establish whether protocatechuic acid can be applied to humans [281]. 

### 9.2. Ellagic Acid

Ellagic acid, an antioxidant, is a naturally occurring phenolic lactone compound that is abundant in strawberries, raspberries, cranberries, and walnuts [289,290]. It polymerizes with gallic acid to form glycoside ellagitannins. The hydrolyzable tannin ellagitannin is readily hydrolyzed in the gastrointestinal tract to produce ellagic acid. Ellagic acid is metabolized by intestinal bacteria into urolithin (Figure 18), which exhibits strong antioxidant activity and enhances the immune system. 

Ellagic acid has been shown to change the composition of the gut microbiota, and is converted to urolithins by gut bacteria, and alleviates oxidative stress and inflammatory diseases in the gastrointestinal tract of animals [292]. 

It also changed the intestinal flora and ameliorated *C. perfringen*-induced enteritis in animal experiments [293]. However, only a few clinical trials have been conducted to date. The ingestion of ellagic acid from foods, such as fermented raspberry juice [294] or *Arbutus unedo* [291] may be beneficial for human health. Ellagic acid was also shown to be effective against cognitive impairment and multiple sclerosis [295,296], suggesting its efficacy in the treatment of neurodegenerative diseases. However, further animal experiments and clinical trials are needed in the future.

## 10. Conclusions

In this review, we introduced compounds that may attenuate some diseases through the involvement of phenolic compounds that exhibit antioxidant activities. Target phenolic compounds must be absorbed to exert their effects, and this requires the cleavage of the sugar of a glycoside. The glycoside is then converted into an aglycone that is subsequently metabolized by phase I and phase II enzymes in the small intestine and liver before circulating in the body. Unabsorbed phenolic compounds undergo biotransformation by intestinal bacteria, after which they are absorbed and circulated in the body. These metabolites exert antioxidant and anti-inflammatory effects.

Although phenolic compounds have been extensively examined in animal and cell culture studies in the last decade, the number of human clinical trials has been insufficient. Research on their effects in humans requires a great deal of effort because detailed planning and massive data collection are required due to large individual differences. Dietary ingredients are safe for consumption, but do not exert immediate effects. Further research on the nutrients present in the daily diet and their beneficial effects is warranted and may provide insights into the prevention or attenuation of diseases. Table 1 summarizes the studies introduced in this review that showed contributions to health. We hope that the efforts and achievements of researchers to date will lead to further advances in this field.

## Figures and Tables

**Figure 1 antioxidants-12-00880-f001:**
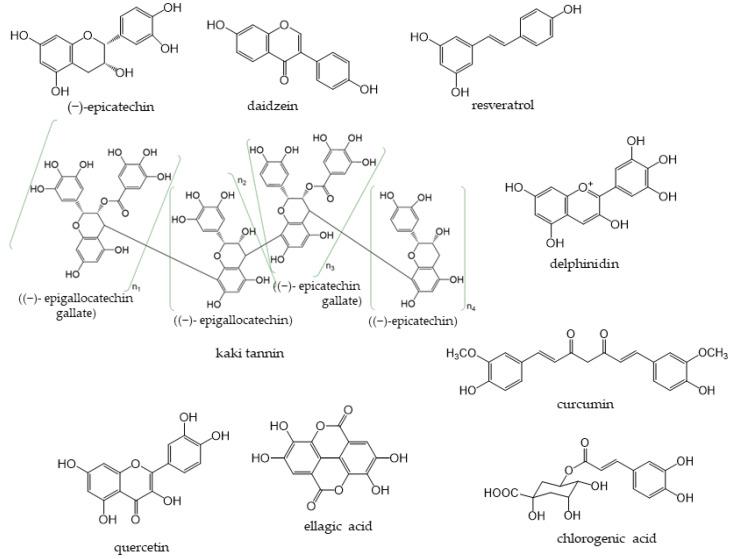
Representative phenolic compounds.

**Figure 2 antioxidants-12-00880-f002:**
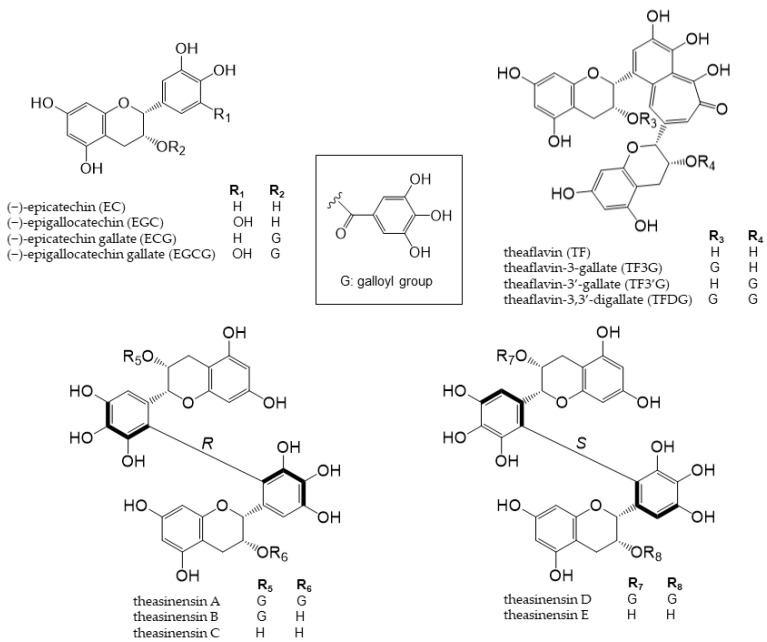
Chemical structures of catechins, theaflavins, and theasinensin A–E.

**Figure 3 antioxidants-12-00880-f003:**
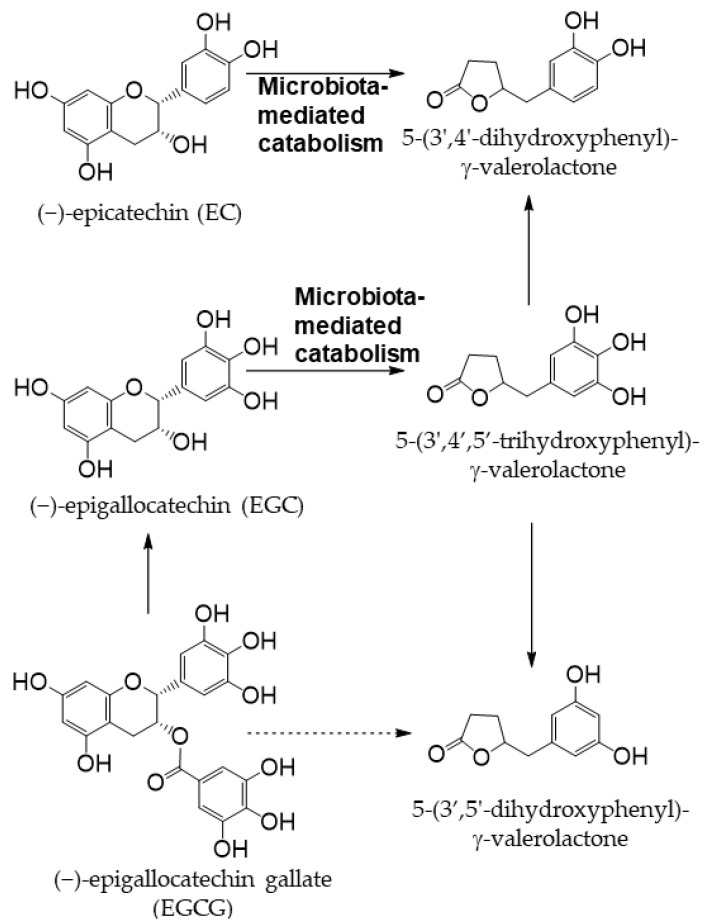
Schematic diagram of the biotransformation of main tea catechins. Modified from [35].

**Figure 4 antioxidants-12-00880-f004:**
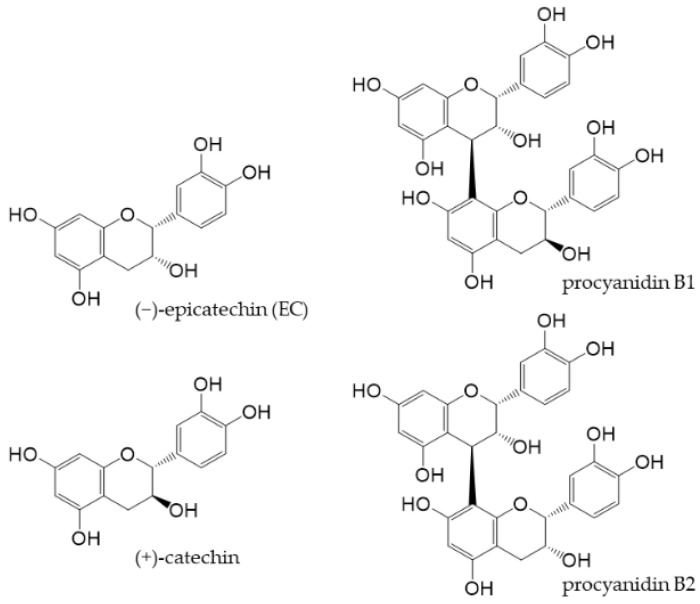
Chemical structures of main flavan-3-ols in cocoa powder.

**Figure 5 antioxidants-12-00880-f005:**
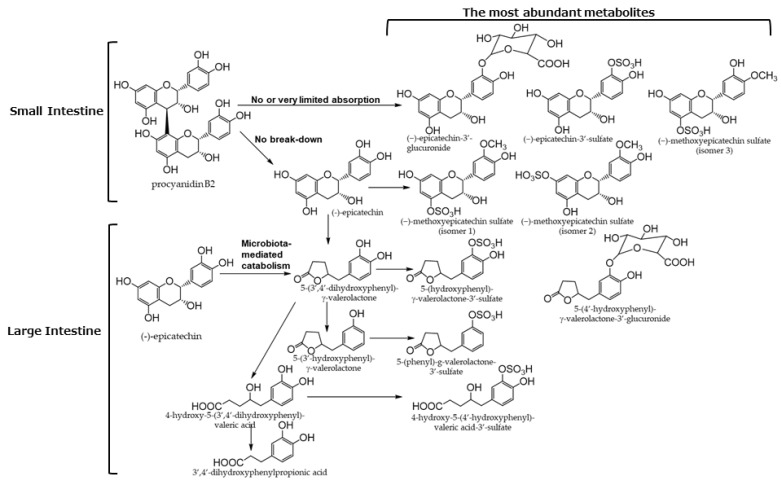
Biotransformation pathways of main flavonols in humans. Quoted from [45].

**Figure 6 antioxidants-12-00880-f006:**
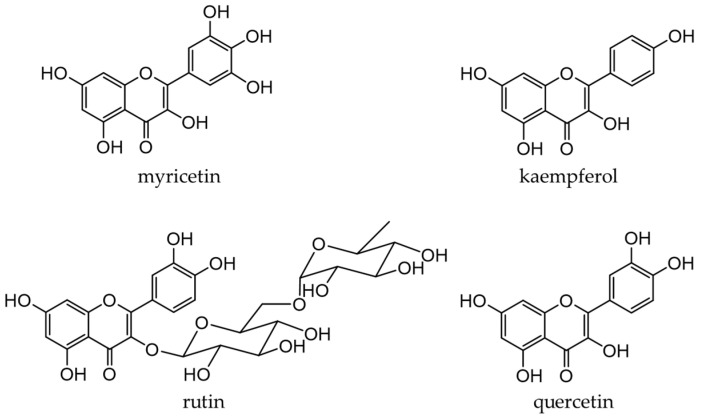
Chemical structures of major flavonols.

**Figure 7 antioxidants-12-00880-f007:**
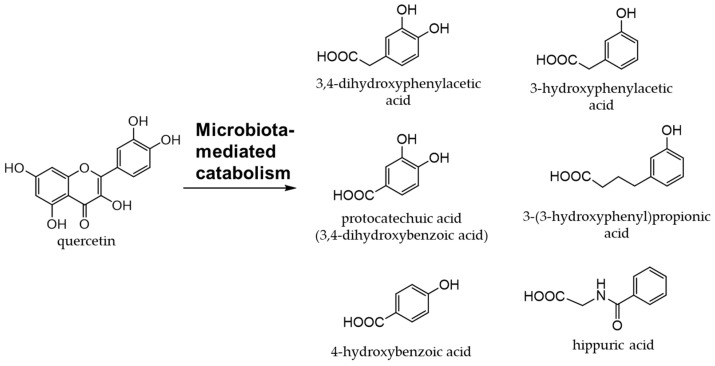
Quercetin catabolites by intestinal bacteria.

**Figure 8 antioxidants-12-00880-f008:**
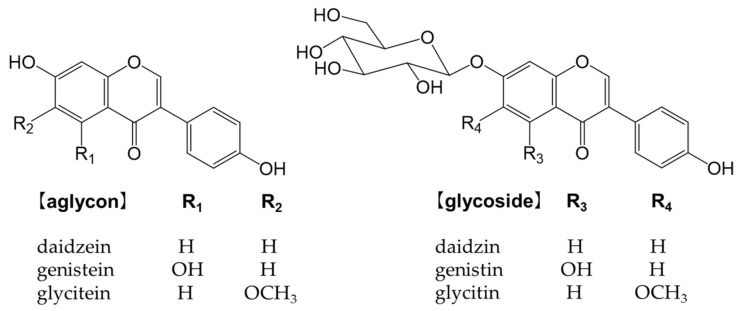
Chemical structure of the main isoflavones and isoflavone glycosides.

**Figure 9 antioxidants-12-00880-f009:**
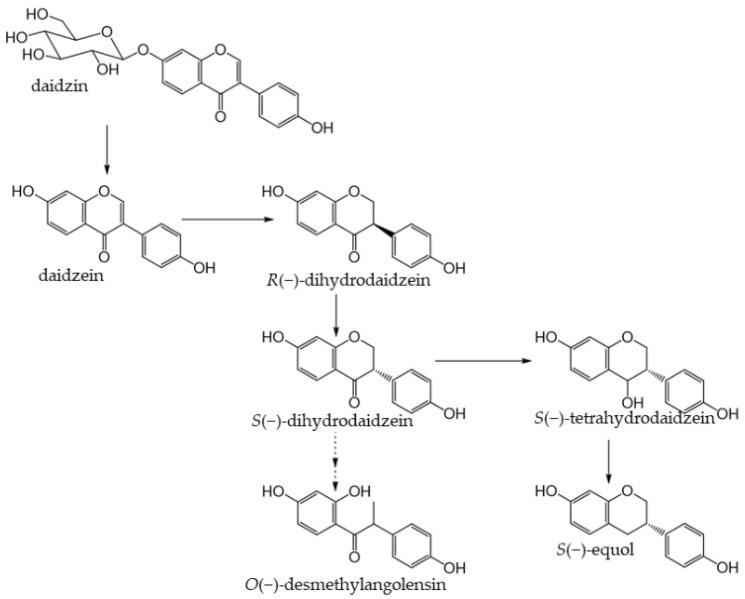
Metabolism of the isoflavone glucoside daidzin by the human gut microbiota and biosynthesis pathway of equol. Modified from [154].

**Figure 10 antioxidants-12-00880-f010:**
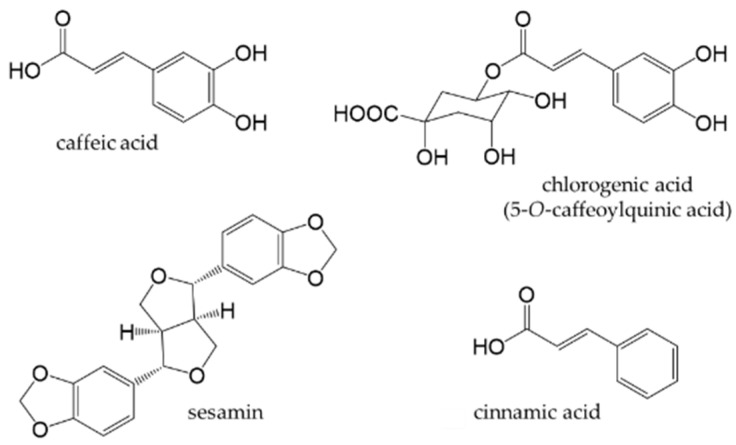
Chemical structures of major phenylpropanoids.

**Figure 11 antioxidants-12-00880-f011:**
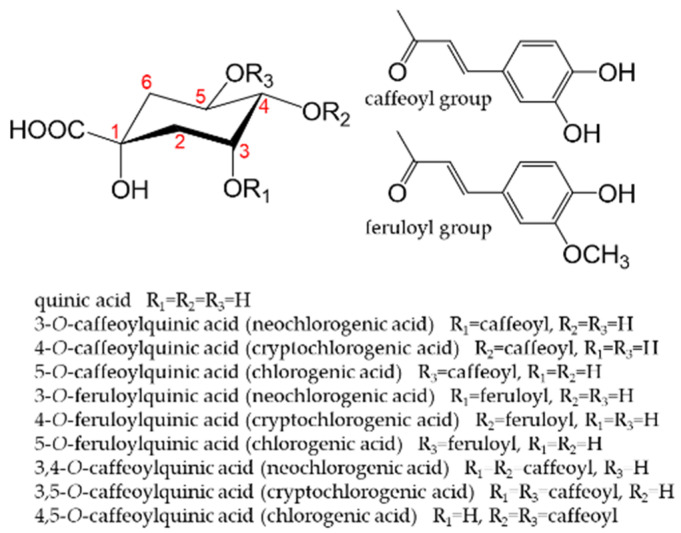
Main chlorogenic acids present in coffee. The numbers in the figure are necessary to indicate where the caffeoyl group or feruoyl group is attached.

**Figure 12 antioxidants-12-00880-f012:**
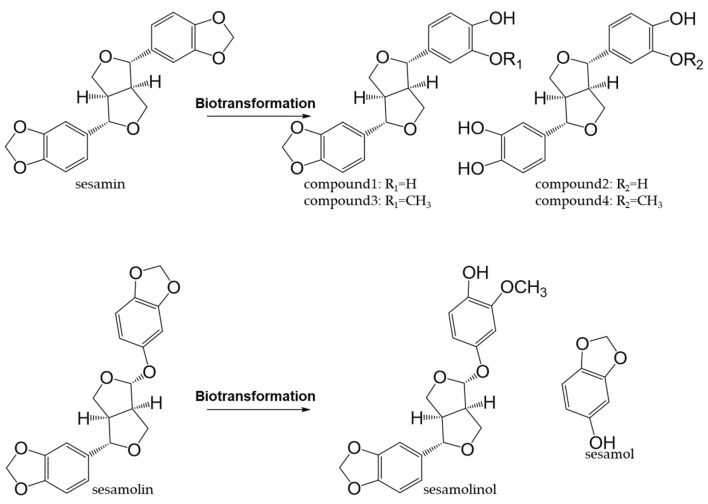
Biotransformation of sesamin and sesamolin.

**Figure 13 antioxidants-12-00880-f013:**
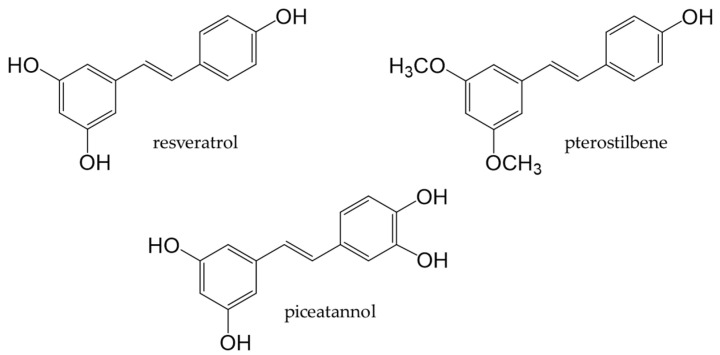
Chemical structures of major stilbenoids.

**Figure 14 antioxidants-12-00880-f014:**
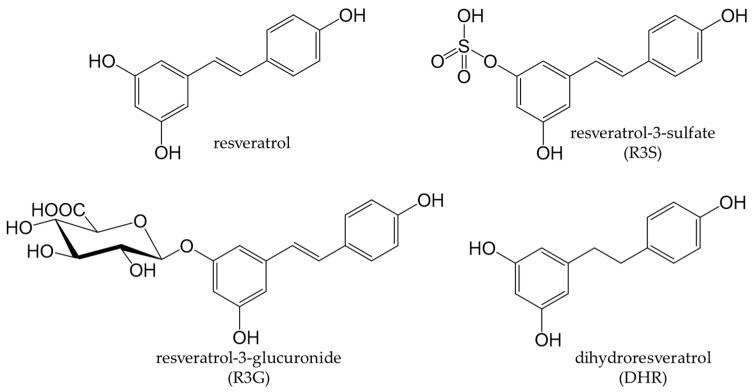
Chemical structures of resveratrol metabolites.

**Figure 15 antioxidants-12-00880-f015:**
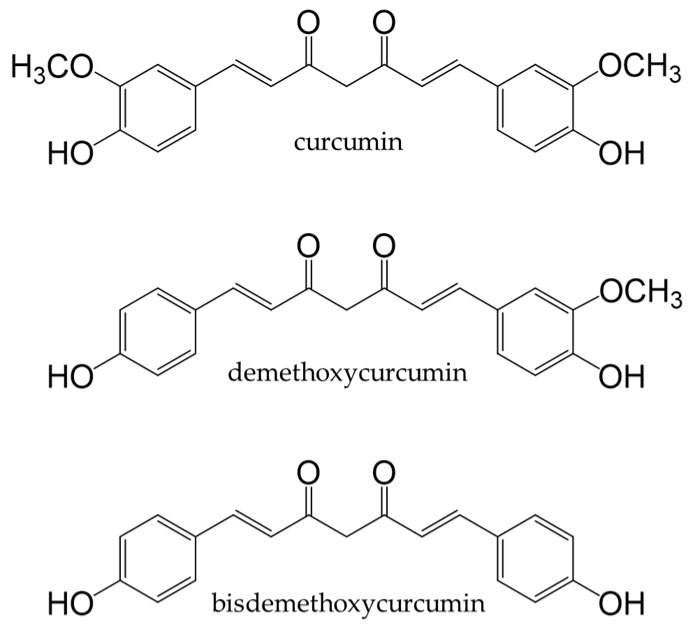
Chemical structures of curcuminoids.

**Figure 16 antioxidants-12-00880-f016:**
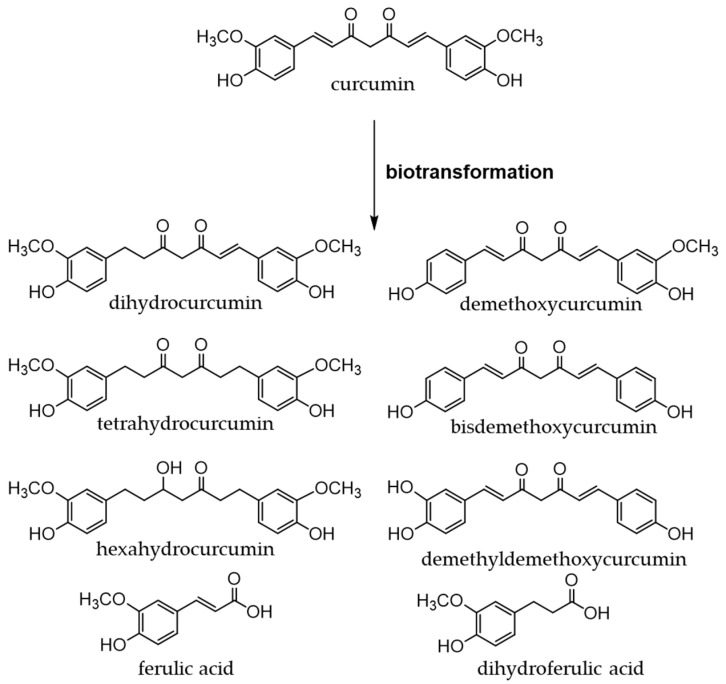
Biotransformation of curcumin.

**Figure 17 antioxidants-12-00880-f017:**
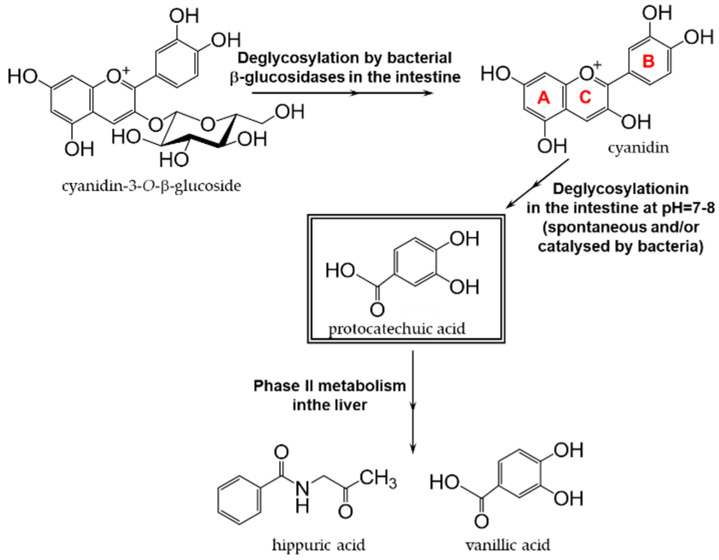
Major metabolic pathway of the anthocyanin cyanidin-*O*-β-glucoside in humans. The presence of intestinal bacteria accelerates the formation of protocatechuic acid through cleavage of the C-ring shown in red letters in figure. Modified from [286].

**Figure 18 antioxidants-12-00880-f018:**
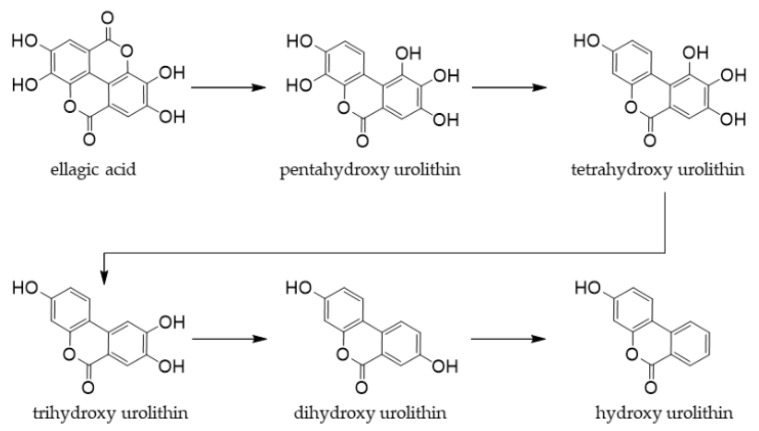
Schematic representation of the production of microbial metabolites from ellagic acid. Modified from [291].

**Table 1 antioxidants-12-00880-t001:** Salutary effects of phenolic compounds.

Dietary Phenolic Compound Source/Compound	Disease	Study Results	Reference(s)
polyphenols	cardiovascular disease	database-linked survey of preclinical trials and clinical trials on polyphenols for the treatment of cardiovascular disease	Behl et al., 2020 [5]
polyphenols	rheumatoid Arthritis	efficacy of polyphenols to mitigate rheumatoid arthritis by inhibiting the MAPK signaling pathway	Behl et al., 2021 [6]
polyphenols	rheumatoid Arthritis	a review of preclinical and clinical data on various pathways involved in rheumatoid arthritis and polyphenols as therapeutic agents	Behl et al., 2022 [7]
plant polyphenols	depression	a review of the chemical, pharmacological, and neurological evidence for the potential of polyphenols in depression	Kabra et al., 2022 [8]
polyphenols	depression	a review of polyphenols that inhibit oxidative stress and inflammation through signaling pathways in depression	Behl et al., 2022 [9]
polyphenols carotenoids	eye disease	a review of the health benefits of polyphenols and carotenoids for the prevention and treatment of age-related eye diseases	Bungau et al., 2019 [10]
quercetin, EC	arteriosclerosis	augmentation of nitric oxide status and attenuation of endothelin-1 concentration in plasma of healthy men	Loke et al., 2008 [47]
cocoa/EC	cardiovascular disease	acute elevations in levels of circulating nitric oxide species, an enhanced flow-mediated vasodilation response of conduit arteries, and an augmented microcirculation	Schroeter et al., 2006 [48]
EC	brain endothelial dysfunction, neurodegenerative disorders	regulated protein expression and gene expression in brain endothelial cells	Corral-Jara et al., 2022 [51]
green tea extracts	alcoholic fatty liver disease	attenuation of triacylglycerol levels in serum and liver and aminotransferase activities in mice	Li et al., 2021 [52]
tea extracts	alcoholic fatty liver disease	prevention of liver steatosis, decrease in oxidative stress and inflammation, modulation of gut microbiota	Li et al., 2021 [54]
green tea	alcoholic fatty liver disease	amelioration of alcoholic liver disease by activation of *Akkermansia muciniphila*	Zhao et al., 2022 [53]
EGCG	non-alcoholic fatty liver disease	inhibited the increase in histological fatty deposits and triglyceride accumulation in the liver induced by high fat diet, improved intestinal dysbiosis, and involved in sirtuin genes	Naito et al., 2020 [55]
concord grape polyphenols	obesity	increase in the growth of *Akkermansia muciniphila* and decrease in the proportion of Firmicutes to Bacteroidetes	Roopchand et al., 2015 [59]
EGCG	ulcerative colitis	the active treatment remission rate was 53.3% (8 of 15) compared with 0% (0 of 4) for placebo	Dryden et al., 2013 [60]
EC	acute and chronic colitis	attenuation of COX-2 expression and increase in cell proliferation, repair of the epithelium by stimulating the expression of EGF	Vasconcelos et al., 2012 [61]
EGCG and piperine	ulcerative colitis	increased bioavailability, decreased colonic histological damage and MDA levels, and increased antioxidant enzyme activity	Brückner et al., 2012 [62]
EGC and ECG	Alzheimer’s disease	attenuation of amyloid-β aggregation, reduced ROS production, less neurotoxicity to neurons	Chen et al., 2020 [65]
EGCG	Alzheimer’s disease	negative regulation of microglial inflammation and neurotoxicity	Zhong et al., 2019 [66]
EGCG	Alzheimer’s disease	activated ERK-and PI3K-mediated pathways in astrocytes and accelerated amyloid-β degradation	Yamamoto et al., 2017 [67]
EGCG	Alzheimer’s disease	inhibition of neuroinflammatory response in microglia, protection from indirect neurotoxicity	Cheng-Chung Wei et al., 2016 [68]
EGCG	Alzheimer’s disease	attenuation of cognitive deficits in APP/PS1 mice	Bao et al., 2020 [69]
EGCG	Parkinson’s disease	modulation of the substantia nigra iron transport protein ferroportin, attenuation of oxidative stress, neuroprotective effects	Xu et al., 2017 [70]
EGCG	Parkinson’s disease	inhibition of substantia nigra neurodegeneration, neuroprotective effect	Sergi 2022 [71]
EGCG	hypoxia-induced neuroinflammation	protection of microglia by disabling the NF-κB pathway and activating the Nrf-2/HO-1 pathway	Kim et al., 2022 [72]
flavanol-enriched cocoa powder	amelioration of intestinal environment	enhanced the abundance of *Lactobacillus* and *Bifidobacterium* species, modulated markers of local gut immunity	Jang et al., 2016 [73]
cocoa flavanols	disorder of the intestinal environment	growth of select gut microflora in humans	Tzounis et al., 2011 [74]
cocoa	disorder of the intestinal environment	improved gut-associated lymphoid tissue function by modulating IgA secretion and gut microbiota	Pérez-Cano et al., 2013 [75]
cocoa	deterioration of the intestinal immune system	differential TLR patterns, attenuation of intestinal IgA secretion and IgA-coating bacteria	Massot-Cladera et al., 2012 [76]
cocoa	diabetes mellitus	amelioration of intestinal flora, barrier integrity, and the inflammatory status of the intestine	Álvarez-Cilleros et al., 2020 [77]
cocoa	inflammation-related colon carcinogenesis	attenuation of NF-κB, pro-inflammatory enzyme expression, and inducible NO synthase expression	Rodríguez-Ramiro et al., 2013 [78]
cocoa flavanols	coronary artery disease	maintenance of normal endothelium-dependent vasodilation	Agostoni C. et al., 2012 [79]
cocoa extract	cardiovascular disease among older adults	lowered risk of total cardiovascular events	Sesso et al., 2022 [80]
cocoa extract	Alzheimer’s disease	modification of the physical structure of amyloid-β oligomers	Dubner et al., 2015 [81]
cocoa extract	Alzheimer’s disease	attenuation of amyloid-β oligomerization	Wang et al., 2014 [82]
cocoa extract	Alzheimer’s disease	neuroprotection by activating the brain-derived neurotrophic factor survival pathway	Cimini et al., 2013 [83]
kaki tannin	metabolic syndrome	strong binding capacity for bile acids	Matsumoto et al., 2011 [87]
kaki tannin	hypercholesterolemia	cholesterol lowering effect and glucose metabolism amelioration by the ability of kaki tannin to bind bile acids	Nishida et al., 2021 [88]
kaki tannin	postprandial hyperglycemia	kaki tannins limited starch digestion and inhibited glucose uptake and transport, thereby alleviating postprandial hyperglycemia	Li et al., 2018 [89]
kaki tannin	disruption of intestinal flora	reshaped fecal gut microbiota	Zhu et al., 2018 [90]
kaki tannin	*Mycobacterium avium* complex (MAC) disease	bacteriostatic effect on MAC, attenuation of pulmonary granuloma formation, suppression of pro-inflammatory cytokine expression	Matsumura et al., 2017 [17]
kaki tannin	ulcerative colitis	decreased disease activity and colonic inflammation, changed microbiota composition and immune response	Kitabatake et al., 2021 [18]
dry persimmon	dyslipidemia	lipid-lowering and antioxidant properties	Gorinstein et al., 1998 [91], Gorinstein et al., 2000 [92]
kaki tannin	hyper-LDL cholesterolemia	attenuation of serum LDL cholesterol levels in humans	Suzuki et al., 2022 [93]
quercetin and isoflavones	osteoporosis	elucidation of metabolic pathways by intestinal microbiota, amelioration of bioavailability	Murota et al., 2018 [95]
quercetin/red onion	obesity and insulin resistance	adipose tissue remodeling	Forney et al., 2018 [118]
quercetin/grape powder	obesity and insulin resistance	prevented macrophage inflammation and adipocyte macrophage-mediated insulin resistance	Overman et al., 2011 [119]
quercetin	kidney disease due to atheroembolism	attenuation of COX-2 induction by stress	Carlsen et al., 2015 [116]
quercetin	obesity-related diseases	antioxidant, anti-inflammatory, and antifibrotic effects on insulin resistance and atherosclerosis	Sato et al., 2020 [123]
quercetin	colitis	rebalanced the pro-inflammatory, anti-inflammatory, and bactericidal function of enteric macrophages	Ju et al., 2018 [120]
quercetin	disruption of intestinal flora	restoration of gut microbiota in mice after antibiotic treatment	Shi et al., 2020 [121]
quercetin	*C. rodentium*-induced colitis	modification of gut microbiota and suppression of proinflammatory cytokines in *Citrobacter rodentium*-induced colitis mice	Lin et al., 2019 [122]
quercetin and rutin	Alzheimer’s disease	anti-amyloidogenic and fibril-disaggregating effects	Jiménez-Aliaga et al., 2011 [124]
quercetin	Alzheimer’s disease	promotion of viability and proliferation of Alzheimer’s disease model cells, increase in expression of sirtuin 1/Nrf2/HO-1 and antioxidant-related enzymes	Yu et al., 2020 [125]
quercetin	Alzheimer’s disease	inhibition of tau protein hyperphosphorylation and oxidative stress, inhibition of PI3K/Akt/GSK3β, MAPK, and NF-κB p65 in a cell line of mouse hippocampal neurons	Jiang et al., 2016 [126]
quercetin	Alzheimer’s disease	inhibition of BACE-1 (Beta-site APP Cleaving Enzyme-1, β-secretase), attenuation of amyloid-β peptide levels	Shimmyo et al., 2008 [127]
quercetin	Alzheimer’s disease	targeted integrated stress response signaling, suppressed amyloid-β (Aβ) production and prevented cognitive impairment in a mouse model	Nakagawa et al., 2019 [128]
quercetin	Parkinson’s disease	activation of the PKD1–Akt cell survival signaling axis, neuroprotective signaling in a dopaminergic neuronal model	Ay et al., 2017 [129]
quercetin	Parkinson’s disease	significant attenuation of rotenone-induced behavioral impairment, augment of autophagy, attenuation of ER stress-induced apoptosis with attenuated oxidative stress	El-Horany et al., 2016 [130]
quercetin with piperine	Parkinson’s disease	attenuation of movement disorders and biochemical and neurotransmitter changes	Sharma et al., 2020 [131]
quercetin with piperine	Parkinson’s disease	significantly amelioration of MPTP-induced behavioral abnormalities in rats, reversal of the abnormal alterations of neurotransmitters in the striatum	Singh et al., 2017 [132]
buckwheat	Hypercholesterolemia, neurodegenerative disease, cancer, inflammation, diabetes, hypertension	buckwheat as a food and its effects on health	Giménez-Bastida et al., 2015 [104]
quercetin, rutin/buckwheat	dyslipidemia, metabolic syndromes,	quercetin reduced obesity due to high-fat diet, rutin, quercetin, and tartary buckwheat shaped specific structures of the intestinal microbiota	Peng et al., 2020 [133]
phenolic compounds/tartary buckwheat	human breast cancer	inhibitory ability of phenolic compounds on breast cancer cell proliferation	Li et al., 2017 [134]
rutin	cancer	regulation of molecular networks and signaling mechanisms in cancer cells by rutin	Perk et al., 2014 [135]
rutin	COVID-19	conformational change upon binding of rutin and SARS-CoV-2 spike protein	Kumari et al., 2022 [136] Rahman et al., 2021 [137]
rutin, quercetin/buckwheat	postprandial rise in blood sugar, diabetes, hypercholesterolemia	the rutin and phenolic compounds contained in buckwheat inhibited the action of digestive enzymes, suppressing the sudden rise in postprandial blood sugar levels and lowering cholesterol	Kreft et al., 2022 [138] Cirkovic Velickovic et al., 2018 [139] Wang et al., 2022 [140] Ikeda et al., 1993 [141] Zhang et al., 2017 [142] Bao et al., 2016 [143]
buckwheat	cardiovascular disease, dyslipidemia	review and meta-analysis on buckwheat and cardiometabolic health	Llanaj et al., 2022 [144]
rutin	neurodegenerative disease	a review of the neuroprotective mechanisms of rutin	Enogieru et al., 2018 [145]
buckwheat	hypercholesterolemia, inflammation, neurodegenerative disease, cancer, diabetes, hypertension, celiac disease	health benefits of buckwheat, potential remedy for diseases	Noreen et al., 2021 [146]
isoflavone	a wide range of hormonal disorders	classification, structure, and occurrence, with their metabolism, biological, and health effects in humans and animals, and their utilization and potential risks	Křížová et al., 2019 [147]
isoflavone and metabolites	cardiovascular diseases, metabolic syndromes, osteoporosis, diabetes, brain-related diseases, etc.	the latest research trends that have shown substantial interest in the biological efficacy of isoflavones in humans and plants, and their related mechanisms	Kim 2021 [148]
isoflavones	some hormone-dependent diseases	effects of isoflavones on chemoprevention of breast cancer, prostate cancer, and cardiovascular osteoporosis and alleviation of osteoporosis and postmenopausal symptoms	Vitale et al., 2013 [156]
S-equol	vasomotor symptoms, osteoporosis, prostate cancer, cardiovascular disease	summary of studies demonstrating effects of isoflavone supplements on menopausal symptoms, bone, prostate cancer, and cardiovascular biomarkers	Jackson et al., 2011 [159]
isoflavone/soybeans	breast, thyroid, and uterus of postmenopausal women	a review of key studies related to soy, with a focus on clinical and epidemiological studies	Messina 2016 [162]
soy protein	blood cholesterol	attenuation of total and LDL cholesterol	Harland et al., 2008 [163]
soy isoflavones	osteoporosis	significant increase in bone density, decrease in urinary deoxypyridinoline, a marker of bone resorption	Wei et al., 2012 [164]
dietary soy	chronic kidney disease	significantly reduced serum creatinine, serum phosphorus, CRP, and proteinuria; no significant change was found in creatinine clearance and glomerular filtration rate	Jing et al., 2016 [165]
fermented soy products	diabetes mellitus, blood pressure, cardiac disorders, and cancer-related issues	attenuation of serum levels of total cholesterol, low-density lipoprotein (LDL), and triglycerides, maintenance of bone health and prevention of osteoporosis and maintenance of normal endothelial function	Jayachandran et al., 2019 [166]
genistein	Alzheimer’s disease	directly targeted amyloid-β and tau to regulate intracellular signaling pathways involved in neuronal death in the brain	Uddin et al., 2019 [168]
soy isoflavones	Alzheimer’s disease	neuroprotective effects on scopolamine-induced memory impairment, enhancement of cholinergic function, suppression of oxidative stress and activation of ERK/CREB/BDNF signaling	Lu et al., 2018 [169]
genistein	Alzheimer’s disease	regulated CAMK4 to regulate tau hyperphosphorylation	Ye et al., 2017 [170]
genistein	Parkinson’s disease	neuroprotective effect on dopaminergic neurons	Arbabi et al., 2016 [171]
genistein	early phases of allergic encephalomyelitis, multiple sclerosis	decreased cell cytotoxicity	Razeghi Jahromi et al., 2014 [172]
sesame	diabetes mellitus, hypercholesterolemia, osteoarthritis, some types of cancer	detailed research on sesame oil contents, health effects, nutraceuticals, oil quality, and value addition strategies	Langyan et al., 2022 [179]
sesame	free radical-related diseases	Nutraceutical, pharmacological, traditional, and industrial value of sesame seeds with respect to bioactive components that have high antioxidant activity	Pathak et al., 2014 [180]
chlorogenic acid	obesity and associated glucose intolerance	attenuation of food intake, elevation of body temperature, increase in heat dissipation and activation of brown adipose tissue	He et al., 2021 [188]
chlorogenic acid	obesity and obesity-related metabolic endotoxemia	suppression of body weight gain, attenuation of relative weight of fat, amelioration of intestinal barrier integrity, prevention of impaired glucose metabolism and endotoxemia, significant alteration of intestinal microbiota composition	Ye et al., 2021 [189]
chlorogenic acid	high-fat diet-induced obesity	attenuation of plasma lipids, alteration of adipose tissue-associated gene expression, reversal of gut microbiota dysbiosis	Wang et al., 2019 [190]
coffee	type 2 diabetes mellitus	attenuation of diabetes risk in humans	Huxley et al., 2009 [191]
coffee	disruption of intestinal flora	increase in the growth of *Bifidobacterium* spp and *Clostridium coccoides*-*Eubacterium rectale* group	Mills et al., 2015 [192]
coffee	disruption of intestinal flora	coffee consumption can selectively improve the growth of probiotic strains, thus exerting a prebiotic effect	Sales et al., 2020 [193]
chlorogenic acid	Parkinson’s disease	activation of Akt/ERK signaling in the mitochondrial intrinsic apoptotic pathway, neuroprotection against MPTP-induced toxicity in a Parkinson’s disease mouse model	Singh et al., 2020 [195]
caffeic acid, chlorogenic acid	Parkinson’s disease	protection of rotenone-induced neurodegeneration of both nigral dopaminergic and enteric neurons, upregulation of metallothionein	Miyazaki et al., 2019 [196]
chlorogenic acid	Parkinson’s disease	attenuation of oxidative stress and neuroinflammation in MPTP-poisoned mice	Singh et al., 2018 [197]
chlorogenic acid	Alzheimer’s disease	attenuation of cognitive deficits in APP/PS1 mice by activation of the mTOR/TFEB signaling pathway	Gao et al., 2020 [198]
sesamin	variety of cardiovascular diseases	attenuation of cardiovascular disease effects on RAS/MAPK, PI3K/AKT, ERK1/2, p38, p53, IL-6, TNFα, and NF-κB signaling networks	Dalibalta et al., 2020 [200]
sesame	climacteric disorder	amelioration of blood lipid, antioxidant, and sex hormone status	Wu et al., 2006 [201]
sesamin	chronic kidney disease	suppression of uremic toxin production by inhibition of bacterial L-tryptophan indole-lyase	Oikawa et al., 2022 [202]
sesamin	disruption of intestinal flora	increase in the adhesive index of probiotics, up-regulation of the adhesive protein (β-cadherin and E-cadherin) expression	Wang et al., 2021 [204]
sesamol	Alzheimer’s disease	attenuation of SCOP-induced cognitive dysfunction via balancing the cholinergic system and reducing neuroinflammation and oxidative stress	Yun et al., 2022 [205]
sesamol	Alzheimer’s disease	attenuation of Alzheimer’s disease-related cognitive impairment and neuroinflammatory response by mediating the gut microbe–SCFA–brain axis	Liu et al., 2021 [206]
sesamin, sesamol	Alzheimer’s disease, Parkinson’s disease, Huntington’s disease	activation of SIRT1/SIRT3/FOXO3a expression, inhibition of BAX (pro-apoptotic protein) and upregulation of BCL-2 (anti-apoptotic protein)	Ruankham et al., 2021 [207]
sesamin	diabetes-induced neurodegenerative diseases	attenuation of microglial activation by high glucose, reduction of inflammatory response and neurotoxicity	Kongtawelert et al., 2022 [208]
sesamin, sesamolin, sesamol	Alzheimer’s disease	sesamin protected against Aβ toxicity by reducing toxic Aβ oligomers, sesamin and sesamolin ameliorated amyloid-β-induced deficits in chemotactic behavior, anti-amyloid-β toxic activity and structure–activity relationship of sesame lignans	Keowkase et al., 2018 [209]
resveratrol	neuroinflammatory disease	prevention of self-destruction of nerve cells	Renaud et al., 2014 [218]
resveratrol/red wine	cardiovascular disease, lung cancer, prostate cancer	effect of red wine on cardiovascular morbidity and mortality	Vidavalur et al., 2006 [219]
red wine	coronary heart disease	inhibition of platelet reactivity by wine (alcohol)	Renaud et al., 1992 [220]
resveratrol	intestinal dysfunction	regulation of intestinal barrier function under immunosuppression	Song et al., 2022 [221]
resveratrol	colitis	activation of metabolism by intestinal microbiota, modification of intestinal microbiota	Yao et al., 2022 [222]
resveratrol	obesity	amelioration of intestinal flora, regulation of lipid metabolism, recovery of intestinal barrier function, amelioration of insulin sensitivity	Wang et al., 2020 [223]
resveratrol	NAFLD	amelioration of insulin resistance, amelioration of intestinal barrier function and intestinal microbiota composition, amelioration of lipid metabolism	Wang et al., 2020 [224]
resveratrol	NAFLD	inhibition of high-fat diet-induced elevation in cannabinoid receptor type 1 (CB1) mRNA expression, inhibition of colonic CB2 mRNA levels, and maintenance of intestinal barrier integrity	Chen et al., 2020 [225]
resveratrol	metabolic and intestinal disease	upregulation of mRNA expression of tight junction and mucin-associated proteins, maintenance of intestinal barrier	Zhang et al., 2021 [226]
resveratrol	metabolic syndrome	regulation of intestinal bacterial composition and metabolism and alteration of steroid metabolism in middle-aged men	Korsholm et al., 2017 [227]
resveratrol	obesity	metabolic activation and amelioration of mitochondrial respiration to muscle fatty acid-derived substrates and caloric restriction-like effect in obese men	Timmers et al., 2011 [228]
resveratrol	cardiovascular disease and a variety of cancers	accumulation of resveratrol in epithelial cells along the aerodigestive tract and presence of potentially active resveratrol metabolites	Walle et al., 2004 [229]
red wine	coronary heart disease	changes in lipid profiles, attenuation of insulin resistance, and decrease in oxidative stress	Castaldo et al., 2019 [230]
wine	obesity	consuming moderate amounts of wine as part of a Mediterranean diet did not promote weight gain or abdominal obesity.	Golan et al., 2017 [231]
resveratrol	pregnancy-related complications	effects of resveratrol on embryogenesis and spermatogenesis mediated by several mechanisms	Novakovic et al., 2022 [232]
grape seed oil	wound	wound-healing properties of the oils of *Vitis vinifera* and *Vaccinium macrocarpon* in animal model	Shivananda Nayak et al., 2011 [233] Al-Warhi et al., 2022 [234]
grape seed oil	ulcerative colitis	oral administration of grape seed oil and grape seed extract showed anti-inflammatory effect and effect on ulcerative colitis	Niknami et al., 2020 [235]
grape seed oil	acute liver injury	grape seed oil suppressed inflammation and protected the liver against acute liver injury caused by oxidative stress	Ismail et al., 2016 [236]
grape seed oil	diabetes mellitus	seed oil of *Vitis davidii* Foex. protected pancreatic β-cells from anti-glucose-induced apoptosis and maintained insulin secretion	Lai et al., 2014 [237]
grape seed oil	erythema of the skin	the application of a cream milky lotion containing grape seed oil was found to ameliorate the skin’s moisture content, sebum content, and erythema	Sharif et al., 2015 [238]
grape seed oil	physiological leg edema in primigravidae	physiological edema in pregnancy was suppressed with foot massage using grape seed oil	Navaee et al., 2020 [239]
grape seed oil	hyperlipidemia	blood triglycerides were suppressed by oral administration of grapeseed oil for 6 weeks	Kaseb et al., 2016 [240]
resveratrol	Alzheimer’s disease	significant attenuation of cytotoxicity of amyloid-β1-42 peptide against SH-SY5Y human neuroblastoma cells, neuroprotective effect	Al-Edresi et al., 2020 [241]
resveratrol	hypoxia, Alzheimer’s disease	prevention of hypoxia-induced upregulation of total amyloid and exosomal amyloid-β by inhibiting CD147	Xie et al., 2019 [242]
resveratrol	Alzheimer’s disease	upregulation of the SIRT1 pathway, induction of cognitive enhancement and neuroprotection against amyloid and tau pathologies	Corpas et al., 2019 [243]
resveratrol	Alzheimer’s disease	activation of AMPK-dependent signaling by resveratrol rescued amyloid-β-mediated neurotoxicity in hNSCs.	Chiang et al., 2018 [244]
resveratrol	Parkinson’s disease	regulation of the MALAT1/miR-129/SNCA signaling pathway	Xia et al., 2019 [245]
resveratrol	Parkinson’s disease	attenuation of MPTP-induced loss of dopaminergic neurons, attenuation of astroglial activation in the nigrostriatal pathway, attenuation of motor dysfunction in MPTP-treated mice	Liu et al., 2019 [246]
resveratrol	Parkinson’s disease	neuroprotective effects of regulation of α-synuclein expression upon loss of miR-214 in Parkinson’s disease	Wang et al., 2015 [247]
resveratrol	Huntington’s disease	improved motor coordination and learning, enhanced expression of mitochondrial-encoded electron transport chain genes in YAC128 mice	Naia et al., 2017 [248]
resveratrol	multiple sclerosis	promoted remyelination effect of resveratrol	Ghaiad et al., 2017 [249]
resveratrol	amyotrophic lateral sclerosis (ALS)	increase in mitochondrial biogenesis in the SOD1(G93A) spinal cord, increase in expression and activation of Sirtuin 1 and AMPK in the ventral spinal cord	Mancuso et al., 2014 [250]
curcumin	cancer	potential of curcumin to influence lipogenic pathways that regulate human cancer cell metabolism	Naeini et al., 2019 [257]
curcumin	various chronic diseases including various types of cancers, diabetes, obesity, cardiovascular, pulmonary, neurological, and autoimmune diseases	Anti-inflammatory activity through the suppression of numerous cells signaling pathways including NF-κB, STAT3, Nrf2, ROS, and COX-2,	Kunnumakkara et al., 2017 [258]
curcumin	cancer	inhibition of activation of Toll-like receptor 4 (TLR4) signaling pathway associated with inflammatory response and cancer progression	Chen et al., 2018 [260]
curcumin	intestinal inflammatory diseases, such as Crohn’s disease, ulcerative colitis, and necrotizing enterocolitis	improved intestinal barrier function, regulated the gut microbiota, exhibited antioxidant and anti-inflammatory effects	Burge et al., 2019 [261]
curcumin	cancer	potent antitumor activity by reversing epigenetic changes associated with oncogene activation and tumor suppressor gene inactivation	Carlos-Reyes et al., 2019 [262]
curcumin	colorectal adenoma	regulation of the Wnt/β-catenin pathway associated with colorectal cancer	Bahrami et al., 2017 [263]
curcumin	colorectal cancer	disruption of tumor growth signaling such as COX-2 enzyme expression, attenuation of NF-kB signaling, suppression of EGFR phosphorylation, inhibition of angiogenesis, and apoptosis of malignant cells	Adiwidjaja et al., 2017 [264]
curcumin	ulcerative colitis	reduced recurrence rates and maintained remission in patients with quiescent ulcerative colitis	Hanai et al., 2006 [265]
curcumin	*Helicobacter pylori*-infected gastritis	although treatment of *H. pylori*-infected patients with curcumin did not alter levels of inflammatory cytokine mRNA expression and had limited anti-bactericidal effect, it improved common symptoms in the patients	Koosirirat et al., 2010 [266]
curcumin	*Helicobacter pylori*-infected gastritis	significant amelioration of dyspeptic symptoms and attenuation of serologic signs of gastric inflammation were observed in *H. pylori*-positive patients with functional dyspepsia despite the lack of eradication of *H. pylori*	Mario et al., 2007 [267]
curcumin	gallstone disease	defense against biliary cholesterol supersaturation by modulating intestinal microbiota and inhibiting intestinal cholesterol absorption	Hong et al., 2022 [268]
curcumin	ulcerative colitis complicated by diabetes mellitus	effectively alleviated colitis in mice with type 2 diabetes by restoring Th17/Treg homeostasis and improving gut microbiota composition	Xiao et al., 2022 [269]
curcumin	intestinal inflammatory diseases	enhancement of the intestinal barrier, attenuation of intestinal apoptosis by suppressing the caspase-3 pathway, reduction in intestinal inflammation by inhibiting the MAPK/NFκB/STAT3 pathway, and amelioration of gut bacteria involved in colitis	Guo et al., 2022 [270]
curcumin	acute myeloid leukemia	promoted responses to cytarabine through modulation of the microbiota, highlighting the importance of enhancing gut integrity in chemoresistance therapy	Liu et al., 2022 [271]
curcumin	irritable bowel syndrome	significant improvement in gastrointestinal symptom rating scale and stress scale indicators	Lopresti et al., 2021 [272]
curcumin	Alzheimer’s disease	effects of curcumin-activated PPARγ on anti-neuroinflammatory and neuroprotective effects in Alzheimer’s disease	Liu et al., 2016 [273]
curcumin	Alzheimer’s disease	blocked amyloid-β aggregation and fibril formation in vitro and in vivo by directly binding curcumin to small beta-amyloid species	Yang et al., 2005 [274]
curcumin	Parkinson’s disease	effective inhibition of the toxic effects of MPP+ on SH-SY5Y cells, greatly attenuating the adverse effects of MPP+ on dopaminergic neurons via upregulation of HSP90	Sang et al., 2018 [275]
curcumin/encapsulated	Huntington’s disease	amelioration of mitochondrial dysfunction and significant enhancement in neuromotor coordination	Sandhir et al., 2014 [276]
curcumin	amyotrophic lateral sclerosis (ALS)	amelioration of aerobic metabolism and oxidative damage, and slowed disease progression	Chico et al., 2018 [277]
curcumin	major depressive disorder	potency to modulate neurotransmitter levels, inflammatory pathways, excitotoxicity, neuroplasticity, hypothalamic–pituitary–adrenal disorders, insulin resistance, oxidative and nitrosative stress, and the endocannabinoid system	Ramaholimihaso et al., 2020 [278]
protocatechuic acid	cancer, hyperlipidemia, diabetes	potential to agent of antioxidant, antibacterial, anticancer, antihyperlipidemic, antidiabetic, and anti-inflammatory	Kakkar et al., 2014 [280]
protocatechuic acid	neurodegenerative disease, tumors, osteoporosis, liver disease, kidney disease, metabolic syndrome	regulation of oxidative stress and inflammatory responses via multiple signaling pathways	Song et al., 2020 [281]
protocatechuic acid/Du-Zhong	chronic hepatotoxicity	attenuation of liver lesions incidence	Hung et al., 2006 [282]
protocatechuic acid	Alzheimer’s disease, Parkinson’s disease	inhibition of β-amyloid plaque accumulation and tau hyperphosphorylation in brain tissue	Krzysztoforska et al., 2019 [286]
protocatechuic acid	NAFLD	regulation of glucose and lipid metabolism, oxidative stress, inflammation, gut microbiota, and metabolites, increase in energy expenditure of brown adipose tissue	Gao et al., 2021 [287]
protocatechuic acid	depression	maintained brain-derived neurotrophic factor levels and modulated oxidative stress responses, cytokine systems, and antioxidant defense systems in mice	Thakare et al., 2021 [288]
ellagic acid	inflammatory disease, neurodegenerative diseases	discovery of a novel bacterial strain capable of converting ellagic acid to isourolithin A with anti-inflammatory, anti-carcinogenic, cardioprotective, and neuroprotective properties	Selma et al., 2017 [292]
ellagic acid	subclinical necrotic enteritis of broiler caused by *Clostridium perfringens*	regulation of jejunal inflammatory signaling pathways TLR/NF-κB and JAK3/STAT6, alleviation of jejunal oxidative stress, inhibition of intestinal barrier damage, prevention of systemic inflammatory response	Tang et al., 2022 [293]
ellagic acid	multiple sclerosis	attenuation of astrogliosis, astrocyte activation, demyelination, neuroinflammation, and axonal damage via NLRP3 inflammasome and pyroptotic pathway	Kiasalari et al., 2021 [295]
ellagic acid	cognitive impairments, long-term potentiation deficits	significant prevention of traumatic brain injury-induced memory impairment and hippocampal long-term potentiation impairment	Farbood et al., 2015 [296]

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
