# Peer review of "Dietary Phenolic Compounds: Their Health Benefits and Association with the Gut Microbiota"

_antioxidants, 2023, doi:10.3390/antiox12040880_

Round 1

Reviewer 1 Report (Previous Reviewer 2)

Most of the issues raised have been addressed. I think that now the manuscript is worth of publication in antioxidants.

Author Response

Point 1: Most of the issues raised have been addressed. I think that now the manuscript is worth of publication in antioxidants.

Response 1: We are so grateful for these positive comments.

Reviewer 2 Report (New Reviewer)

As I see, this is a revised manuscript (yellow backgrounded text/). No novelty is provided, nor special aspects or a relevant presentation of the manuscript. The authors must do serious work in cleaning, structuring and correct developing their paper. Please see my main suggestions.

Introduction is too poor for the topic. And poor referenced (first reference appears only in L46). As being a Review, each statement must be referenced properly, as follows:

L37. after ..”dependent on its chemical structure” I suggest adding as reference https://doi.org/10.37358/RC.19.9.7497

Figure 1 must be closer inserted in the text where it is mentioned for the first time (after the first paragraph of Introduction).

L42. Different types of diets (I suggest checking and referring to https://doi.org/10.1007/s12035-020-02065-3 andhttps://doi.org/10.1016/j.lfs.2020.118661 ) must be mentioned, detailing the role of polyphenols in each of it and the implication in therapy of different diseases (cardiovascular https://doi.org/10.1016/j.biopha.2020.110714, rheumatoid arthritis https://doi.org/10.3390/molecules26216570 https://doi.org/10.1080/10408398.2021.1924613 ; depression DOI: 10.3389/fphar.2022.1046599 and https://doi.org/10.1016/j.biopha.2021.112545 ; eye diseases https://doi.org/10.1155/2019/9783429   etc.

L61-65. Aim of the study. As the topic is not a new one, in the last paragraph of Introduction, please highlight/detail the special aspects/novelty that your study brings to the field, and what differentiate this paper from other in the same topic. Why have you chosen this topic? 

Some figures are blurred. Please provide better quality ones.

How have the authors selected the references regarding this topic as many recent relevant references have not been provided and many important aspects have not been discussed? A short 2. Methodology of literature selection must be done. Also, discuss here the impact of the topic, by key words. Were logical Boolean operators used between the search terms to link them to the topic? Have you made a Web of Science search regarding the impact of the topic in the literature and to see if it is needed to approach this polyphenols topic? As this paper looks, I am sure that the answer is No.

L248. Onions are NOT the only dietary source of flavonols. Check the literature and improve subsection 3.1.

When referring to dietary source, at each type/class of active compunds, more sources must be provided.

Subsection 6.2.1. For grape composition and therapeutic effects, I suggest https://doi.org/10.3390/life13010178

Self-citations 75-77. What is the link to the topic of this paper? Please remove. Also, please clean the References section, as many papers have nothing to do with the content.

Author Response

Reviewer 3 Report (New Reviewer)

Evaluation of the manuscript „Dietary Polyphenols: Their Health Benefits and Association with the Gut Microbiota” sent to Antioxidants (MDPI) by Yoko Matsumura and co-authors.

The paper is interesting, fits into the journal’s scope and covers wide area of knowledge. After minor improvements could be published.

Line 13: Polyphenols are phytochemicals that include flavonoids, lignans, stilbenoids, and anthocyanidins. ?? To my knowledge anthocyanidins should be regarded as flavonoids, in line with flavonols, flavones, flavanols, flavanones, and isoflavones. To me you should add “tannisns”.

Query: for example in the paragraph about “coffee” you discuss the content and properties of phenolic acids that are not polyphenols but are phenolics together with polyphenolics. Why did you decide to name the work about polyphenolics and not about phenolic compounds?

I know that there are several attitudes as for those phytochemicals division. But make sure the reader that you know, for instance, that phenolic acids are not polyphenols.

Round 2

Reviewer 2 Report (New Reviewer)

The authors responded to my suggestions.

Author Response

Thank you so much for reviewing.

This manuscript is a resubmission of an earlier submission. The following is a list of the peer review reports and author responses from that submission.

Round 1

Reviewer 1 Report

Although the authors introduced different types of dietary polyphenols and their benifit roles, regarding the topoic "Association between Antioxidant Compounds and Gut Microbiota", it is quite weak. The authors need to strengthen how gut microbes participate in polyphenol action, and list the prospects of future research on this field.

Reviewer 2 Report

The paper by Matsumura et al. is a survey of the literature on the effects of dietary polyphenols on gut microbiota and/or on various diseases. Although all the compounds taken into account show antioxidant activity, little seems to be known on the connection between the antioxidant activity and the biological effects of the compounds. Indeed, the Authors repeatedly state that further research on this subject is required. The review could be a useful source of information on the therapeutic use of dietary polyphenols, but the association between antioxidant compounds and Gut Microbiota to give heath benefits is not sufficiently clarified.

For this reason I suggest to change the title of the review, which could be misleading.

Moreover, the conclusions shoul be improved and revised . espcially the statement “compounds that may improve some diseases through the involvement of polyphenols that possess antioxidant activities. To exert their effects, the target (?)  polyphenols must be absorbed”.

Minor points:

Abstract, line 14:” Polyphenols are phytochemicals that include flavonoids and lignans and so on”. Please improve the sentence with more details, removing “and so on”

Line 68: What do the Authors mean by “intramolecular hydroxyl groups”?

Line 109 the sentence: “however, it has negligible antioxidant properties” says the opposite of what was stated a few lines above

Line 143: flavanols are most abundantly present in EC?.

Line 149: Cocoa powder alters the metabolites of gut microbiota: this is not clear to me

Figure 5 the name of compounds with methoxy groups should be corrected, e.g. . (-)-methoxyepicatechin sulfate

Line 245 “absorbed isoflavone aglycones are mainly metabolized to derivatives (?) of glucuronides and”…It is not clear to me

Line 279 from the statement “Phenylpropanoids, also called lignoids, are compounds in which multiple Phenylpropanes…” it seems that they include only polimers, however they include also monomers

Line 283: resveratrol is a type of phenylpropanoid?

Figure 11: feruloyl not feruoyl

Figure 13 and line 349: Dihydro not dihydroxyresveratrol

Reviewer 3 Report

This review paper (Health Benefits of Dietary Polyphenols: Association between Antioxidant Compounds and Gut Microbiota) is interesting and much worthy of investigation. Overall paper is reporting some interesting facts. But, I have some major concerns regarding this manuscript.

1- Author should include some more explanation regarding the need of polyphenols and its sources in introduction section. A normal food is rich in polyphenols. However, it's difficult to know how much of that benefit is actually due to the specific plant compound or to all the nutrients, fiber, and other phytochemicals also found in those foods. Author should put data related to it.

2- Line 43-45; In recent years, Author should refer some investigations briefly that has been performed to investigate the mechanism of polyphenols derived from various foods. Just overview would be enough.

3- The Figures 1-17; representing polyphenols can be used as such. Either it was made by author or just got from another paper. Probably it need to get permission if its not associated with creative common license. Author may use software for designing formulas.

4- In most of the polyphenols authors relied on basic information, even it lacks the mechanism in detail especially related to the gut microbiota. It will be more appropriate if author add some mechanism base figures for better understanding. Its just an option but for adding more data regarding the  mechanism is compulsory in such kind of review paper.

5-  It’s a common concept that supplemental polyphenols may raise the risk of cancer and reduce the synthesis of hormones. Author should also include such kind of limitations in his article.

6- I will suggest author to add a table, having different kind of polyphenols and their effects. Please check the topic, available at Science direct (https://www.sciencedirect.com/topics/agricultural-and-biological-sciences/polyphenol).

7- Author should add more subheadings in each discussed polyphenol e.g., dietary source, metabolism, nutritional significance. It will enhance the readability of article.

8- According to recent advancement in science most of the scientists are focusing on “gut–brain axis-based health benefits” of polyphenols. I didn’t see such kind of information in article. Author is suggested to add.

9-  Polyphenols have effect of anti-inflammatory, antiaging, cardioprotective, and anticancer. Author is suggested to add opinions  of recent studies regarding the relation of antioxidants and these mentioned effects.

10-  Overall, the language is understandable in manuscript, but sill it needs a thoroughly check for grammatical type of mistakes.